# Exploring the impact of mobility and selection on stone tool recycling behaviors through agent-based simulation

Emily Coco *

Center for the Study of Human Origins, Department of Anthropology, New York University, New York, New York, United States of America

* ec3307@nyu.edu

## Abstract

Recycling behaviors are becoming increasingly recognized as important parts of the production and use of stone tools in the Paleolithic. Yet, there are still no well-defined expectations for how recycling affects the appearance of the archaeological record across landscapes. Using an agent-based model of recycling in surface contexts, this study looks how the archaeological record changes under different conditions of recycling frequency, occupational intensity, mobility, and artifact selection. The simulations also show that while an increased number of recycled artifacts across a landscape does indicate the occurrence of more scavenging and recycling behaviors generally, the location of large numbers of recycled artifacts is not necessarily where the scavenging itself happened. This is particularly true when mobility patterns mean each foraging group spend more time moving around the landscape. The results of the simulations also demonstrate that recycled artifacts are typically those that have been exposed longer in surface contexts, confirming hypothesized relationships between recycling and exposure. In addition to these findings, the recycling simulation shows how archaeological record formation due to recycling behaviors is affected by mobility strategies and selection preferences. While only a simplified model of recycling behaviors, the results of this simulations give us insight into how to better interpret recycling behaviors from the archaeological record, specifically demonstrating the importance of contextualizing the occurrence of recycled artifacts on a wider landscape-level scale.

## Introduction

Stone tool recycling has occurred since the earliest manufacture of stone tools. Schiffer [1] was among the first to discuss recycling as a cultural process affecting the formation of the archaeological record. Subsequent archaeologists have elaborated on his definition to distinguish between recycling as a functional change and recycling as waste utilization [2]. Here, recycling is defined as "secondary recycling" where stone tools are scavenged and then reworked, requiring some period of discard between episodes of use and typically involving a change in function [2–4]. This definition of recycling separates it from typical discussions of recycling as an aspect of tool maintenance in the context of technological efficiency and curation [3]. Curation

**Data Availability Statement:** All files for the model code and subsequent analyses are available at https://github.com/cocoemily/recycling-Java.

**Funding:** EC was supported by the National Science Foundation under Grant No. 2133751 and by the Leakey Foundation. The funders had no role

in study design, data collection and analysis, decision to publish, or preparation of the manuscript.

**Competing interests:** The authors have declared that no competing interests exist.

is often defined as a characteristic of stone toolkits where tools are produced in anticipation of future use or for transport between locations [5] resulting in large investments in maintenance [6]. Other conceptualizations of curation refer to it at the level of individual artifacts as a ratio between possible and realized utility [7, 8]. Curation as defined in these ways implied conceptualization of stone tool use within a single population. Conversely, recycling as defined in this paper considers stone tool use on archaeological and geological time scales.

The most reliable indication of this type of recycling is evidence of knapping after some sort of surface alteration, such as a patina or rock varnish, has formed, resulting in flake scars with differing degrees of alteration [4, 9–13]. In the absence of this clear "double patina", there is a lack of consensus on how to most accurately identify recycling evidence [3, 14]. Other methods for identifying recycling have focused on showing that stone tools had multiple functional purposes [15–21]. However, identifying a change in function in not always straightforward and this change does not always necessitate discard of the artifact first [22]. Additionally, there are complications of analyzing the stone tool record if it has been impacted by behaviors such as recycling [22–24]. Given these issues, recycling of stone tools has been relatively understudied in the history of archaeological research.

Fortunately, today, archaeologists are increasingly recognizing the importance of recycling behaviors in the production and use of stone tools in the Paleolithic. One example is the identification of recycling as an important component of the Acheulean technological systems in the Levant [e.g. 21, 25–28]. Multiple studies of sites in Israel have demonstrated that lithic recycling is a distinct aspect of stone tool production in Acheulean assemblages. Lithic recycling in these contexts often have two main trajectories: 1) recycling bifaces into cores for the production of flakes, and 2) the production of small flakes from cores-on-flakes or flaked flakes, many of which are patinated [25]. Another example of extensive study of recycling is the work of Vaquero and colleagues at Abric Romaní where they identify patinated and burned flakes subsequently used for tool manufacture [4, 29]. Here, recycling occurs in a Middle Paleolithic context, joining many other examples of recycling behaviors in the Middle Paleolithic [9, 11, 16, 19, 30–32]. At Abric Romaní, Vaquero and colleagues use intra-site spatial analyses and refitting to demonstrate that recycling behaviors lead to movement of artifacts within a site [4, 29].

Despite these advances in trying to understand recycling at specific sites, there are not well-defined expectations for what recycling should look like across the archaeological record. Current research focuses on identifying and characterizing recycling behaviors within layers or within sites [4, 9, 10, 12, 13, 15, 18, 31]. Yet, recycling frequently involves activities offsite, such as scavenging of artifacts from other locations [11, 15, 18, 28, 30, 33, 34]. This means that recycling behaviors do not only affect the archaeological record at the location where recycled objects are found, but potentially also at offsite locations where other assemblages are disturbed by scavenging behaviors. This suggests that recycling behaviors are best contextualized as a property of regional land use. Some of models for land use incorporate recycling as an important motivator for mobility decisions, but these models assume that reuse and reoccupation is exclusively an additive process, resulting in more material at a given location [35, 36]. It is plausible that recycling is a removal process at some sites as items are scavenged for use elsewhere [37, 38]. Archaeological findings have shown that recycling behaviors can lead to the removal of particular artifact shapes from assemblages [18, 39].

Current archaeological proxies do not consider how recycling can rewrite the patterns we rely on for documenting behaviors beyond recycling (but see [40, 41]). For example, archaeologists often use raw material transport as an indicator of mobility; if an assemblage contains raw materials from a distant source, we assume long distance movements of the makers of that assemblage. However, if sites act as new "sources" for recyclable materials, then the raw material composition of assemblages reflect time-averaged movements that were not necessarily

long distance [40–42]. In such a situation recycling is a niche constructing process that prioritizes reoccupation of previously created sites [36, 43, 44]. Barrett's [41] simulations of different raw material acquisition scenarios demonstrate that the distance decay relationship between raw material proportions and distance to geological source disappears when stone tool users prioritize scavenging materials from previously discarded assemblages. This is an excellent example of how recycling behaviors have implications on the behavioral inferences that can be drawn from archaeological proxies.

Few other models of the formation of the archaeological record explicitly allow for scavenging of previously discarded objects despite suggestions by some that stone tool recycling could cause deviations from modeled patterns [45]. To understand how recycling behaviors effect archaeological record formation, I developed an agent-based model to simulate varying probabilities of recycling behaviors by mobile agents. The simulation models surface sites, which are important locations for investigating spatial patterning of behavior in the archaeological record [46, 47]. Furthermore, there is ethnographic and archaeological evidence for scavenging artifacts from surface deposits [34, 48, 49], making this site type a good case study for simulating recycling behaviors. In the model agents move across a gridded landscape and interact with nodules and flakes that can be scavenged and knapped. Using the agent-based model, it is possible to produce sets of expectations for the appearance of a recycled archaeological record through time under different conditions of scavenging frequency, occupation intensity, mobility, and artifact selection.

This model tests an exposure model for recycling in surface deposits; when deposits that have been exposed for longer this will facilitate more opportunities for their discovery resulting in more frequent recycling behaviors [35, 50]. In surface records characterized by geological stability through time, exposure corresponds to age of artifacts, so under the exposure model, older artifacts in surface contexts are more likely to show indications of recycling. Taking this hypothesis one step further, it is possible that older assemblages would also have higher proportions of recycled artifacts. While seemingly logical, this assumes that recycling happens *in situ*, with scavenging, (re)use, and discard of artifacts all occurring in the same location. In this paper, this assumption is not made; instead the impact of recycling behaviors is investigated on a landscape scale under varied mobility conditions that facilitate different degrees of local discard [37, 47]. Specifically, I test whether the number of recycled objects in an assemblage is indicative of the behaviors that occurred at that location or if any such relationships are erased by the repeated scavenging and movement of artifacts around the landscape. For example, one hypothesis to test is whether recycled objects appear more often in locations that have experienced more scavenging events. This model also allows for investigating whether established archaeological proxies, like cortex ratios for studying mobility patterns, become less reliable when recycling behaviors are occurring.

The recycling of stone tools is a powerful force which can repeatedly rewrite archaeological patterns during the formation of the archaeological record. As archaeologists continue to develop methods for tackling the interpretation of a dataset that is the emergent outcome of many individual actions through time [24, 41, 51, 52], it is important that we add recycling of the archaeological record to our understanding of this emergence.

## Model description

The simulation used for this study simulates simplified stone tool scavenging, manufacture, and discard behaviors on a gridded landscape. The simulation was coded in Java 16.0.2 [53]. Model description following the ODD protocol [54–56] is available in S1 Text. This model builds on the methodologies of multiple previously published models, including one by the

author [23 and sources therein] simulating recycling behaviors, and Davies and colleagues' FMODEL simulating artifact discard [37].

This model differs from the previous recycling model published by the author and colleagues [23] in a few major ways. Firstly, the new model does not simulate geological events. This means that the simulated landscape is more akin to surface accumulations where all artifacts are exposed indefinitely upon discard. Secondly, the model presented here more explicitly simulates mobility of agents using Lévy walks (described below). Additionally, the new model incorporates selection criteria for scavenging artifacts based on size and object type. For this reason, the updated model has two types of objects: flakes and nodules. Following the methodology used in the FMODEL [37], nodules are icosahedra (20-sided objects) comprised of flakes that can be removed, and flakes are objects that can be retouched. The size of flakes dictates how many flakes each nodule has. For example, if flakes have a maximum size of 1, then each nodule is comprised of 20 flakes. In this study, we experiment with different flakes sizes, so in some cases, some flakes are size 2, which would mean fewer flakes per nodule.

The modeled world is a 10 x 10 grid, where each grid square represents a location that agents can visit and perform lithic scavenging and reduction behaviors. Agents, representing groups that produce and use stone tools, enter the modeled environment sequentially; when one agent moves out of the landscape, another is subsequently placed on a random grid square. Each iteration of the model is run until all agents have moved through the landscape. Model runs were conducted with 100 agents and 200 agents to simulation different occupational intensity.

At each timestep, the agent performs four behaviors (Fig 1). Firstly, the agent scavenges available artifacts from the current grid square they are occupying. The agent will then either make new flakes by removing them from nodules or retouch previously created flakes. The probability of each of these behaviors is determined by a probability of blank creation and its inverse, respectively. Once the agent has performed its lithic production/reduction, it will discard artifacts and then move to a new position on the landscape.

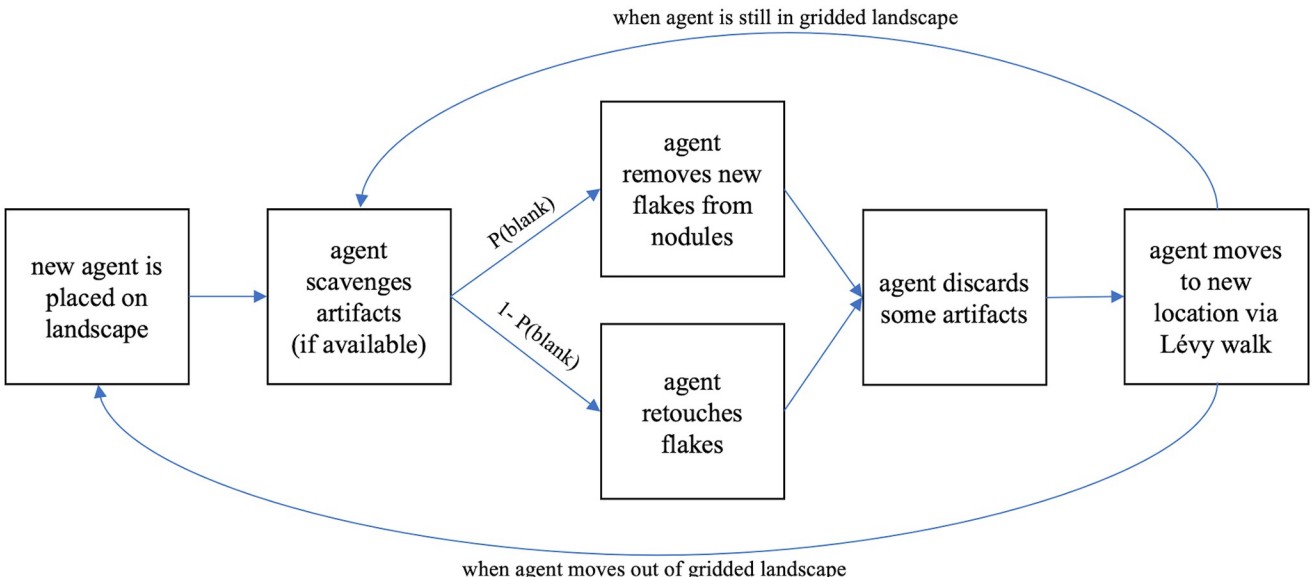

**Fig 1. Model run steps.** Schematic of procedures at each time step of model run.

Recycling behaviors are explored in two different technological contexts. The number of unique technologies available for each agent to use is determined by an overlap parameter. For half of the model experiments, there were only two technology types simulated (overlap is 1); each agent was randomly assigned a technology type of 1 or 2. For the other half of model experiments, each agent had a unique technology type identifier (overlap is 2). Although in most archaeological cases not every group or population will have completely independent stone tool technologies, including this "many technologies" scenario allows for exploring whether the ease of identifying recycling implements has an impact on the patterns produced by the model. In the model context, recycled artifacts are defined as those where the technology used to remove a flake from a nodule differs from the latest technology used to knap or retouch the flake. When each agent has a unique technology type, this recycling indication will occur more frequently.

The rest of model behavior is controlled by multiple parameters that defined the types of objects agents interacted with, probabilities of different actions (i.e., blank creation or scavenging), selection criteria, and movement (see full description in ODD). Each combination of parameters was run 50 times to account for stochastic variation. The code for the model and all subsequent analyses is available at https://github.com/cocoemily/recycling-Java.

## Simulating scavenging and recycling

In this model, scavenging and recycling are simulated by allowing agents to pick up previously discarded objects from the landscape. Agents can interact only with the assemblage of artifacts that is at their current location. If there are objects in that assemblage, agents will pick up artifacts with varying probability (see explanation of scavenging probability parameter in ODD).

Following the scavenging of artifacts, agents will either manufacture new flakes or retouch previously made flakes depending on a user-supplied probability; these actions are performed on randomly chosen nodules or flakes, respectively. This means that the agent does not preferentially recycle previously discarded objects.

This model does not rely solely on scavenged artifacts for raw materials. If an agent does not have any objects in hand at any timestep of the model, then the agent is automatically resupplied with new nodules. In this way, the model simulates local raw material availability because the agents do not need to travel to a particular source to gather new raw materials.

## Simulating selection behaviors

Lithic artifact selection is simulated in two ways governed by three parameters: 1) a preference for either flakes or nodules, 2) a preference or no preference for particular flake sizes, and 3) whether these preferences are strict or not. These parameters come into play when agents are scavenging artifacts from assemblages and when agents are discarding artifacts that they cannot carry with them on their next move. For scavenging, artifacts that match the selection criteria will be prioritized for collection. When selection is strict, agents will not collect any artifacts that do not meet the selection criteria. When selection is not strict, agents will first collect those artifacts matching the selection criteria, followed by further collection of other artifacts randomly as needed. For discard, agents will choose to carry (not drop) artifacts that match the selection criteria.

## Simulating agent movement

Agents are initially randomly placed on the landscape; after initial placement, agents perform Lévy walks within the gridded landscape until they step beyond the limits of the grid. This model uses a similar methodology to the one outlined for FMODEL [37]. The direction that

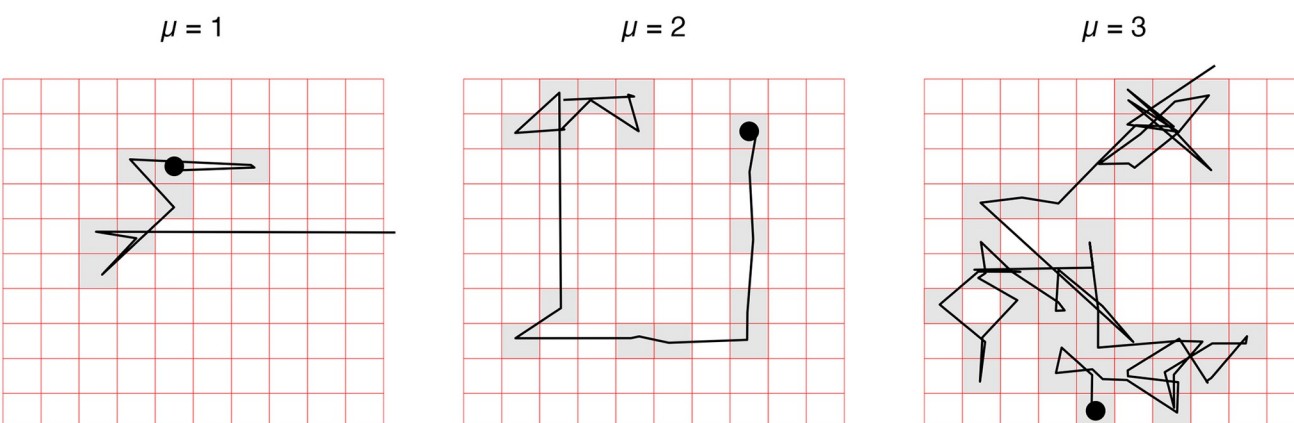

**Fig 2. Comparison of example paths produced under different values of μ.** The black dot represents the starting location of the agent. The grey squares are those occupied by the agent.

the agent faces is chosen randomly from degrees 0 through 360. In Lévy walks, the probability of a step length (l) is determined by Eq 1:

$$P(l) = l^{-\mu} \tag{1}$$

where $P(l)$ is the probability of a step length, $l$ [57]. Reorganizing this equation allows for randomly drawing step lengths via Eq 2:

$$l = P(l)^{-1/\mu} \tag{2}$$

where the probability of the step length $P(l)$ is generated as a random number.

Using this equation allows for varying how long agents spend inside the gridded landscape by making their movement more or less tortuous [37, 41, 58] as is shown in Fig 2. When μ is greater than or equal to 3, agents are more likely to take shorter steps, leading to the agent spending more time inside the modeled landscape as its path frequently intersects and doubles back on itself. As μ approaches a value of 1, there is a higher probability of longer step lengths, meaning the agent is more likely to exit the landscape quickly in a more linear path. Once an agent moves outside of the window of observation, that agent and all the artifacts it is carrying are removed from the model and a new agent is placed on the landscape.

Modeling agent movement in this way simulates different aspects of mobility without needing to define specific points on the landscape as residential bases or logistical camps [41]. The relative frequencies of long-distance and short-distance steps mimics the intensity with which groups occupy a landscape as well as the redundancy in the coverage of that landscape [58]. For example, in a logistical mobility system, areas around a base camp may occupied more intensely with large amounts of small step lengths, whereas logistical forays will be comprised of more longer step lengths [58]. Conversely, in a residential mobility scheme place use if more equal across a landscape.

## Output variables

At different points during a model run, data were outputted summarizing the model state at three different levels: individual objects, individual grid squares, and the entire landscape.

For individual objects, data were outputted after the middle time step of a model run and the final time step of a model run. For each object, the following data were recorded:

1. Whether the object was a flake or a nodule;

2. The model year at the time when the artifact was first dropped by an agent;

3. The stage of the artifact, which for flakes increases by 1 every time a retouch action is taken;

4. The total number of agents that have retouched or produced blanks from the object;

5. The technology type of the agent that initially produced the flake or first reduced the nodule (first technology type);

6. The technology type of the last agent to retouch or produce blanks from the object at the current model year (last technology type);

7. Whether the object has been recycled, which is true when the first technology type does not match the current last technology type.

For each grid square in the landscape, data were recorded every 300 time steps. This produced snapshots of the landscape at 100 different points during a model run. For each grid square, the following data were recorded:

1. A count of all nodule objects;

2. A count of all flake objects;

3. The cortex ratio of the square (see below for explanation);

4. A recycling incidence ratio, which is the number of recycled objects divided by the total object count (both flakes and nodules);

5. The number of discard events, which is increased by the number of flakes/nodules dropped every time discard behaviors happen in the square;

6. The number of scavenging events, which is increased by 1 for every object scavenged from the square;

7. The number of encounters/occupations, which is increased by 1 every time an agent occupies the square;

8. The number of retouch events, which is increased by 1 for every flake retouched

At the model level, data were recorded at every time step to understand the overall character of the landscape throughout each model run. For the model, the following data were collected:

1. The number of scavenging events that occurred during that model year;

2. The number of discard events that occurred during that model year;

3. The number of recycled objects created during that model year;

4. The number of flakes retouched during that model year;

5. The number of blanks produced during that model year;

6. The total number of recycled items currently on the landscape;

7. The total number of assemblages, or grid squares containing at least one object;

8. The total number of encounters summed for all grid squares;

9. The total number of discard events summed for all grid squares;

10. The total number of retouch events summed for all grid squares;

11.  The overall cortex ratio of all objects on the landscape;

12.  The overall recycling incidence ratio, which is the total number of recycled objects in the landscape divided by the total number of objects in the landscape

The calculation of cortex ratios followed the methodology used in the FMODEL [37]. Nodules are initialized as completely cortical with a starting surface area of 11091.8 square units; the size of each flake (either 1 or 2) determines the amount of nodule cortex that flake makes up, either one twentieth or two twentieths of nodule surface area respectively. Nodules are given a starting volume of 100000 cubed units; flake volume is either 4% or 8% of the total nodule volume depending on the flake size (either 1 or 2, respectively). These volume percentages allow for a portion of the nodule's volume to remain after all flakes are removed. Expected and observed surface areas are then calculated based on the number of nodules and the number of flakes in an assemblage.

## Model analysis and results

### Artifact level findings

Camilli and Ebert [35] posited that exposed deposits should be more subject to recycling because exposure facilitates discovery of artifacts by making them visible. In a context of pure surface deposits, such as the one produced in the model, all artifacts are forever exposed following discard. This means that it is the length of exposure that should facilitate discovery with deposits created earlier in a model run being more likely to be scavenged.

I investigated whether recycled artifacts have longer exposure times than non-recycled artifacts by looking at when artifacts are first discarded during a model run. This is tracked in the model by a variable (year of first discard) that stores the model year of when an artifact enters the discard record for the first time. The year of first discard variable for recycled artifacts and for non-recycled artifacts were compared via one-sided Wilcoxon rank-sum tests. Most parameter sets produced recycled objects with older year of first discard dates compared to non-recycled objects. This means that artifacts that are eventually recycled entered the discard record earlier during a model run than artifacts that remained unrecycled.

Under some parameters, recycled artifacts do not have significantly older year of first discard compared to non-recycled artifacts. This occurs when agents cannot scavenge any artifacts that do not exactly match their selection criteria (strict selection). When agents are more limited in the types of objects they are willing to scavenge, it is possible they are forced to scavenge younger artifacts that fit their selection preferences. Alternatively, agents carry out artifacts with the longest exposure times because they preferentially discard objects that do not match their selection criteria.

Using binomial logistic regressions on the significance of the Wilcoxon tests, it is possible to determine how each model parameter affects the likelihood of recycled artifacts being older (and exposed for longer) compared to non-recycled artifacts. Regressions were performed via the MASS package [59]. The results of the regression (Fig 3) demonstrate that when each agent spends more time in the landscape (higher values of μ) recycled artifacts are more likely to occur on longer exposed artifacts. More agents occupying the landscape during model run also increases the likelihood of older objects being recycled. Interestingly, there is no parameter that makes the reverse pattern of recycled artifacts having younger first discard dates significantly more likely.

When considering all artifacts within assemblages, the proportion of recycled artifacts in an assemblage is typically higher in those assemblages that have predominantly older artifacts (Fig 4). This relationship was investigated by looking at the skew (calculated with the moments

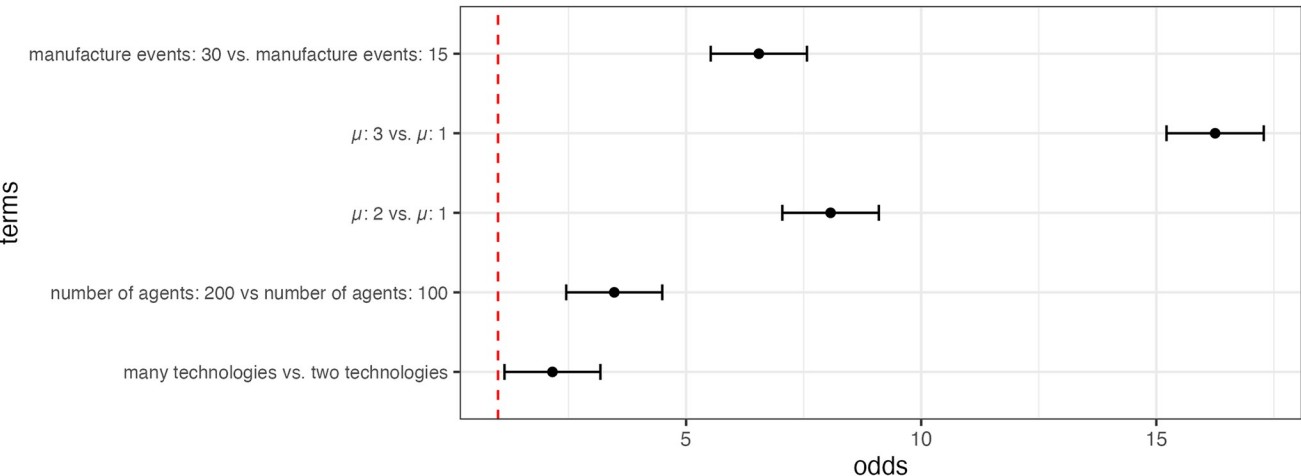

**Fig 3. Odds ratio estimates for each model parameter on the likelihood of recycled objects having older year of first discard dates.** Only significant effects plotted.

package [60]) of the distribution of the year of initial discard variable for artifacts within each assemblage. A negative skew of this distribution means an assemblage dominated by older artifacts; a positive skew would indicate more younger artifacts. The negative slopes in Fig 3 show increased recycling incidence values are associated with assemblages comprised of predominately older artifacts. This relationship is relatively consistent when recycling behaviors are frequent. This suggests that there will be more recycled artifacts in the oldest assemblages when recycling happens frequently throughout time.

This pattern of high recycling incidence values in older assemblages is also relatively consistent across all mobility and selection parameters (S1 Fig). Weakly positive relationships between recycling incidence and skew only occur when selection is strict, which is consistent with the above findings. High recycling incidence values also occur in assemblages with predominately younger artifacts when agents have a non-strict flake preference and no size preference, but only when each agent spends lots of time on the landscape (μ is high). Given the very low R-squared values for each relationship, it is possible that this positive correlation is simply a product of the variation in the relationship between recycling incidence and skew of the year of first discard distribution.

These results demonstrate that in most cases recycled artifacts are more likely to have been exposed longer than non-recycled artifacts in the archaeological record. This supports the hypothesized relationship between recycling and length of exposure time in surface deposits. It also suggests that surface deposits which have been accumulating for long periods of time are likely to be targeted for scavenging artifacts in the context of recycling behaviors.

## Model level findings

The primary variable of interest for this model is the ratio of recycled artifacts that end up in assemblages. For this paper, I refer to this ratio as "recycling incidence." Recycling behaviors are primarily controlled by the combination of scavenging probability and blank creation probability in the model. Scavenging probability dictates how likely agents are to pick up previously discarded artifacts. Blank creation probability controls the frequency of removing new flakes from nodules. The inverse of blank creation probability determines frequency of

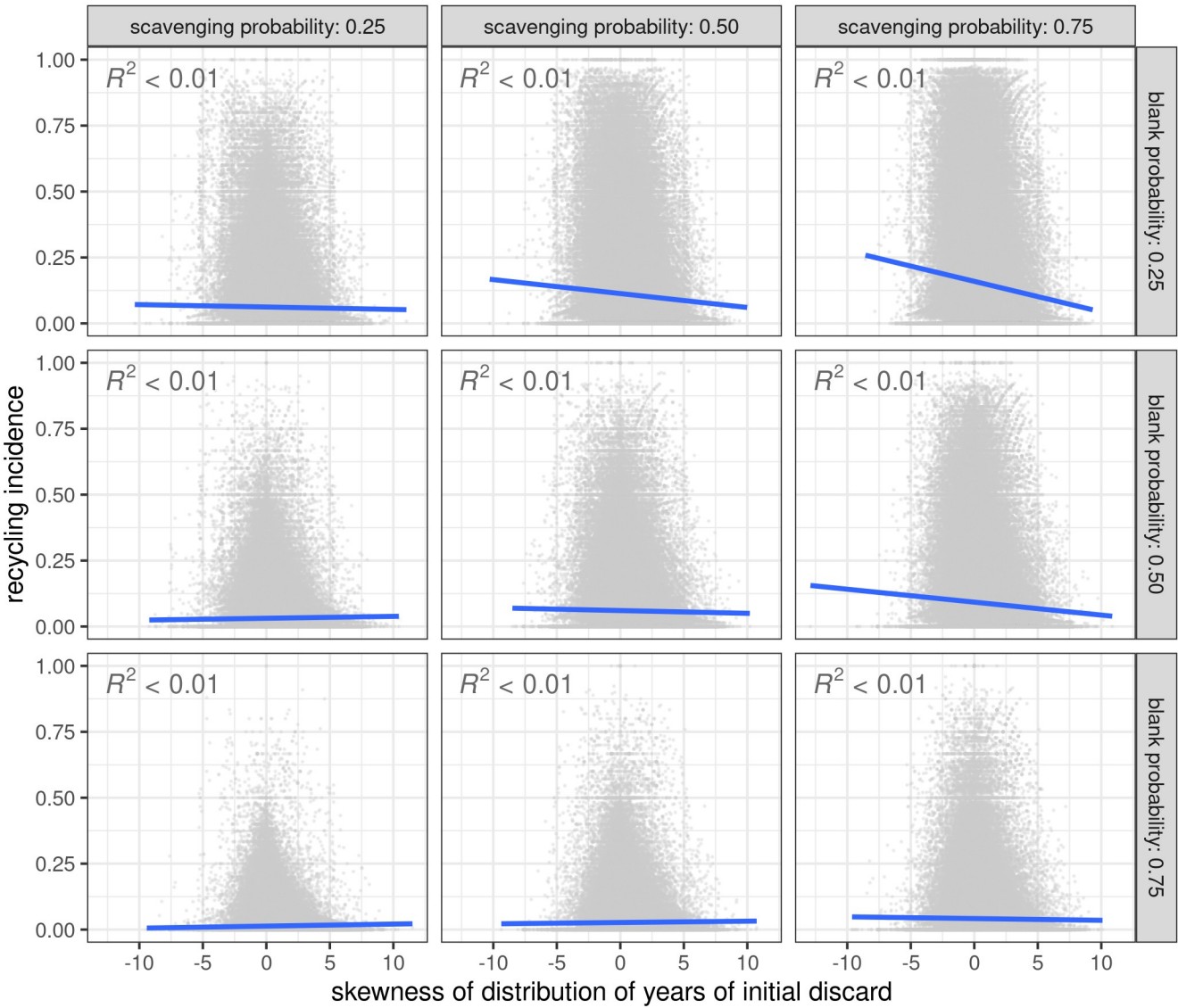

**Fig 4. Relationship between recycling incidence (proportion of recycled artifacts in a grid square) and skewness of distribution of each artifact's year of initial discard.** Results shown for each grid square when agents have one of two technology types (overlap = 1) and only 100 agents occupy the landscape during model run. Negative skew indicates artifacts in the assemblage are first discarded earlier in model run. Positive skew indicates artifacts are discarded later in model run. Linear relationship shown by dark blue line. R squared values given in upper left corner of each panel.

retouching previously created flakes. Since recycled artifacts in this model are most frequently flakes (S2 Fig), a high occurrence of recycling behaviors requires frequent scavenging of flakes (high scavenging probability) and then retouching them (low blank creation probability).

When looking at average values of recycling incidence across the modeled landscape at the end of model run (Fig 5), it is clear that recycling incidence values are highest when recycling is happening frequently (i.e., scavenging probability is high and blank probability is low). This pattern holds whether agents are using one of two technology types during a model run or if each agent has a unique technology type (many technology types). When more blanks are being created, the variation in resulting recycling incidence values is reduced. This is likely

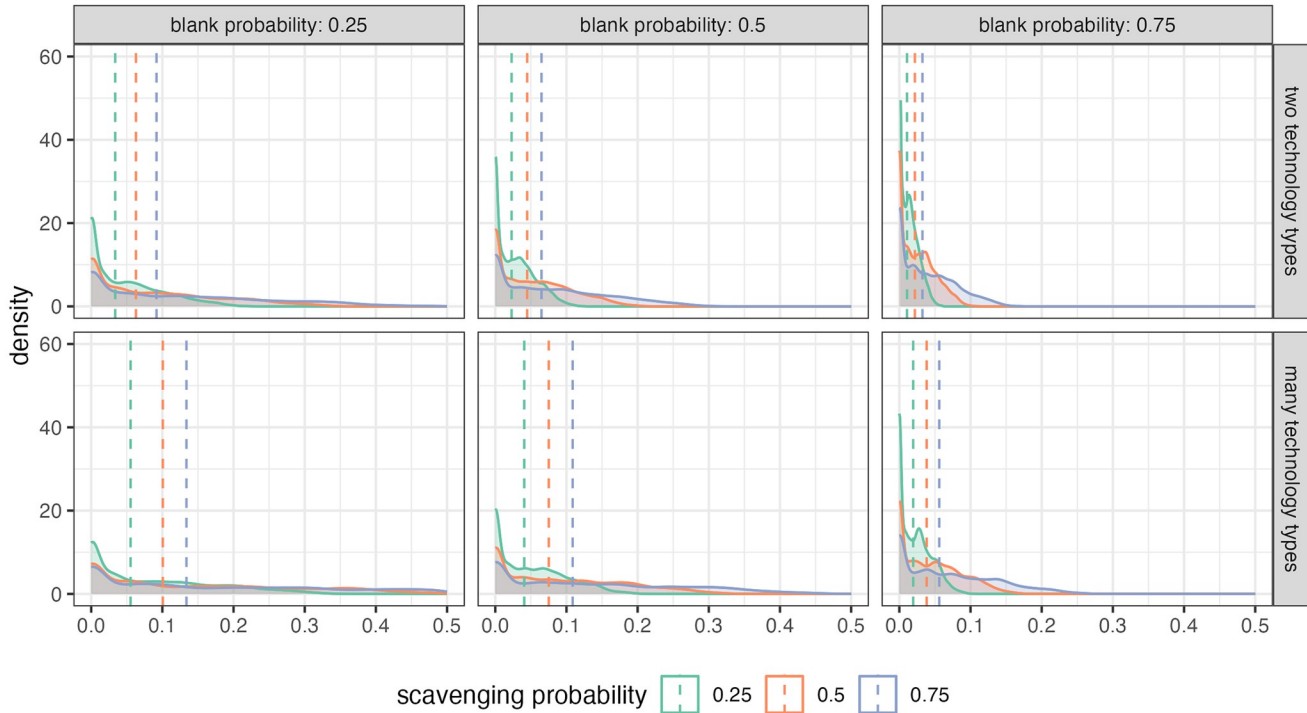

**Fig 5. Density distributions of average recycling incidence values for each model run.** Curves show variation in recycling incidence (RI) values. More diffuse curves show a wider range of RI values produced when scavenging probability is high. Left-skewed curves indicate more low RI values, which occur when scavenging probability is low. Dotted lines show the mean value of the distributions.

because increased blank production increases the total number of artifacts found in the discard record (S3 Fig), increasing the denominator of the recycling incidence ratio.

Recycling incidence values increase with the number of agents that occupy the landscape during model run (S4 Fig) because more agents mean more recycling opportunities. Increased number of agents appears to mitigate the differences in recycling incidence caused by agents making many small steps (higher values of μ): when more agents occupy the landscape, the average recycling incidence values are more tightly clustered for μ values of 2 and 3.

In terms of agent selection preferences, non-strict selection results in higher recycling incidence values compared to strict selection (S4 Fig) because under strict selection agents are more limited in the artifacts they are allowed to scavenging. This necessarily reduces the number of recycled objects an agent can produce. Interestingly, when selection is not strict, there is no significant difference in the recycling incidence values produced under flake preference or nodule preferences (Wilcoxon signed rank test, z = 9915064, p = 0.576). Introducing a size preference reduces recycling incidence values for the same reason as strict selection: size preference means fewer objects are allowed to be scavenged.

To examine model behavior over time, I looked at the trends over time in following outputs: recycling incidence, number of identifiably recycled objects created, number of blanks created, number of scavenging events, number of retouch events, and number of discard events. For each timestep of a model run data was collected on how many of the above events occurred. This allows for an investigation of how frequently behavioral events occur during a model run and how these frequencies were affected by model parameters. Because models are run until every agent has walked through the landscape, every model differs slightly in how

many timesteps it requires; this causes the variation seen at the left end of the trend lines in all the graphs in this section.

As would be expected, the trends for recycling incidence values and number of identifiably recycled objects produced at each timestep are similar (Fig 6). Recycled objects are identifiable in the model only if the first and last technology types used to manufacture the object have different signatures. The first technology type corresponds to that of the agent who initially removed the flake from its nodule or that of the agent who first reduced the nodule. The last technology type refers to that of the agent who performed the last retouch action on a flake or removed the most recent flake from a nodule. When each agent has a unique technology type signature (overlap is 2), there are more opportunities for the first and last technology types to be different. When agents have one of two different technologies (overlap is 1), there will be some cases when recycling occurs, meaning two different agents interact with the same object at different points in time, but the technological signature does not change. As a result, more identifiably recycled objects are produced when each agent has a unique technology type (many technology types) (Fig 6B).

To model different mobility scenarios, the value of μ is used to determine how long an agent occupies the landscape by determining the likelihood of longer and shorter step lengths [37]. When μ is high, agents will take more short steps, causing them to spend more time within the modeled landscape. When μ is low, agents are more likely to take long steps and exit the modeled landscape quickly. When each agent spends more time on landscape, recycling incidence and number of identifiably recycled objects produced are higher. Additionally, there appears to be some threshold of in the likelihood of short steps (as demonstrated by difference in trend lines for μ = 1 versus μ = 2 and μ = 3), above which recycling signatures are quite similar (Fig 6C & 6D).

The frequency of recycling behaviors mediates the effects of mobility on the resulting recycling incidence values (Fig 7). Lower amounts of recycling (low scavenging probability and high blank probability) result in trend lines that are similar across mobility scenarios. This is also true regardless of the agent's selection criteria (S5 Fig). These results suggests that when recycling is happening relatively rarely, the mobility strategy and selection criteria of a group will not significantly impact the signal of recycling as measured by proportion of recycled objects in the archaeological record.

There are slightly different trends for recycling incidence and number of identifiably recycled objects produced when agents preferentially scavenge flakes compared to when they preferentially scavenge nodules (Fig 6E & 6F). A flake preference results in *lower* proportions of recycled artifacts (recycling incidence) at each time step despite producing roughly the same number of identifiably recycled objects per time step. This is because flake preference scenarios result in fewer recycled objects being discarded (see below) compared to nodule preference scenarios (S2 Fig). Additionally, overall more artifacts end up in the discard record when flakes are preferred for scavenging (S3 Fig). As a result, the proportion of recycled artifacts is lower.

When agents prefer flakes of particular sizes (size preference is true), more identifiably recycled objects produced per timestep, but this does not result in similarly higher recycling incidence values. This is contrary to what one might expect, since more selection criteria reduces the set of objects that an agent is willing scavenge. However, a size preference results in overall more artifacts on the landscape compared to no size preference (S3 Fig). This means that there would be more artifacts available for the agents to scavenge, increasing the number of recycled objects that could be created. A strict size preference, on the other hand, decreases both recycling incidence and the number of recycled objects produced per timestep. This is consistent with the limitations on scavenging opportunities imposed by strict selection criteria.

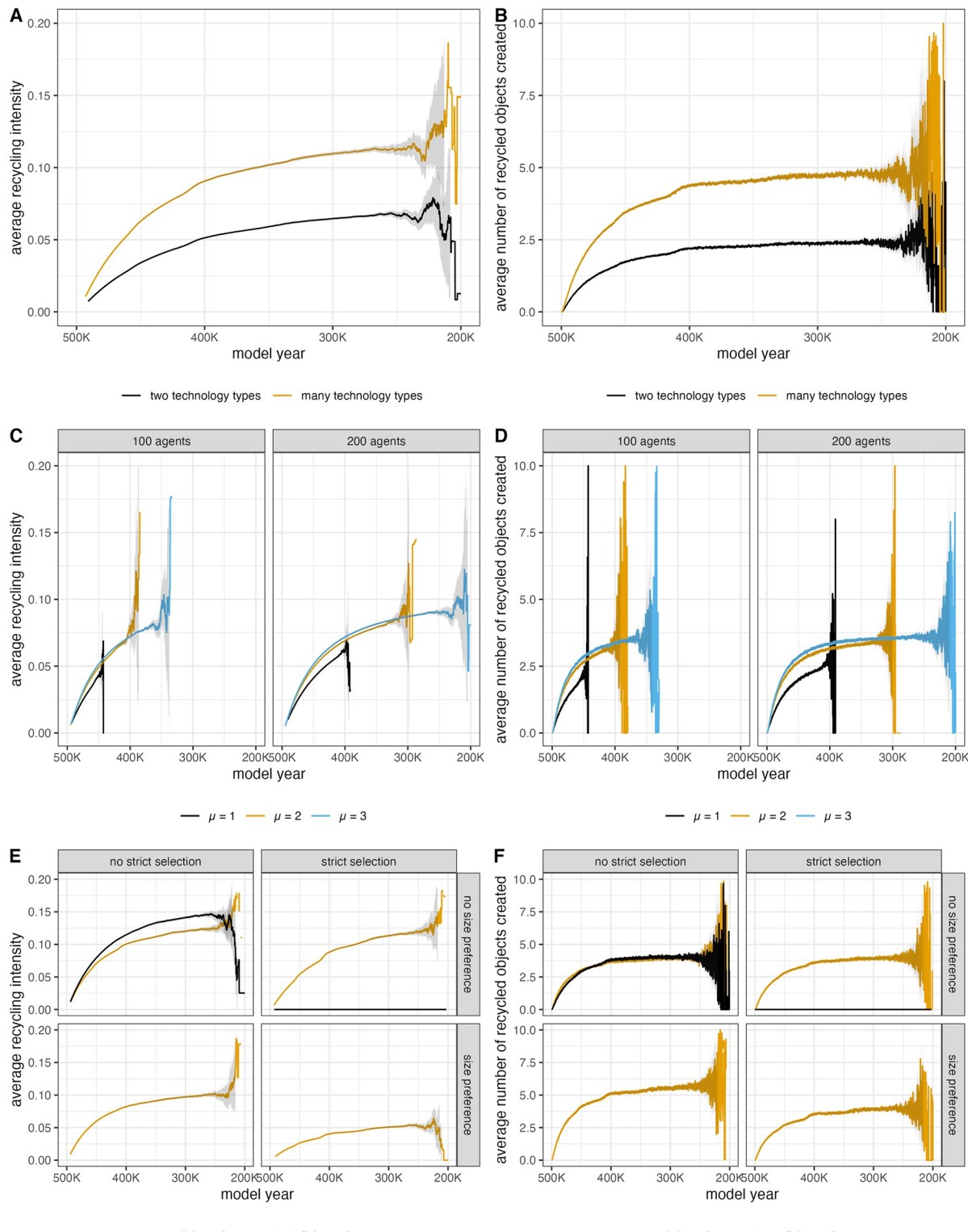

**Fig 6. Average recycling incidence values and number of identifiably recycled artifacts produced at each model year (timestep) of a model run.** Trend lines are compared by overlap parameter (A & B), by μ parameters (C & D), and by selection parameters (E & F). For the selection parameter graphs, a size preference only applies to flakes, so there are no nodule preference trend lines for the size preference panels. Variation on the right side of the graph produced because some model runs last for more "model years" in order to simulate every agent passing through the landscape.

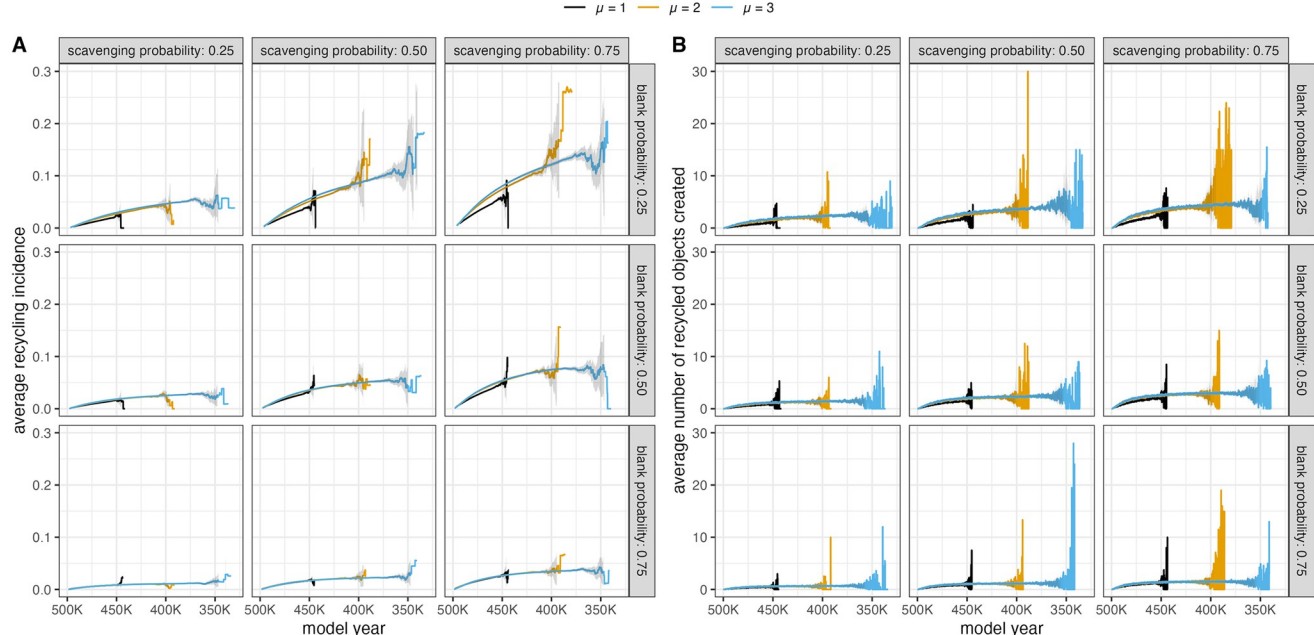

**Fig 7. Average recycling incidence value (A) and number of recycled objects created (B) per time step for different values of scavenging probability and blank probability, which control the frequency of recycling.** High scavenging probability and low blank probability results in more recycling. Low scavenging probability and high blank probability results in less recycling. Variation on the right side of the graph produced because some model runs last for more "model years" in order to simulate every agent passing through the landscape. Trend lines displayed for model runs where agents have one of two technology types (overlap = 1) and 100 agents will occupy the landscape during a model run.

When agents may only scavenge nodules (a strict nodule preference scenario), no identifiably recycled objects are produced (S2 Fig). This results in the flat nodule preference line in Fig 6E & 6F. An agent will never scavenge flakes with a strict nodule preference, making the production of recycled flakes impossible. The fact that no recycled nodules end up in the discard record suggests that agents either rarely discard nodules under these conditions or that any discarded nodules are always scavenged by other agents. This is supported by the very small number of nodules that end up in the discard record when there is a strict nodule preference (S3 Fig).

For the behavioral outputs (i.e., number of blanks created, number of discards, number of scavenging events, number of retouches), whether or not each agent has a unique technology type does not impact the trends (S6 Fig). However, different mobility scenarios do. When each agent spends more time on the landscape (high μ value), agents create more blanks, and scavenge, discard, and retouch more artifacts on average (Fig 8). This effect may be in part due a burn-in effect [61] since one timestep is used to place a new agent into the landscape and agents are initialized with no objects. Higher μ values result in longer agent activity periods, therefore reducing the number of timesteps in a model run used to initialize a new agent.

In terms of selection scenarios (Fig 9), agents scavenge and discard more artifacts per timestep when they preferentially scavenge nodules. Conversely, agents will retouch fewer artifacts when they have a nodule preference. When selection criteria are strict, a nodule preference always results in agents scavenging, discarding, and retouching fewer artifacts. The patterns for blank creation are more unique. The number of blanks agents produce at each timestep is essentially the same regardless of selection criteria. The inclusion of a size preference appears

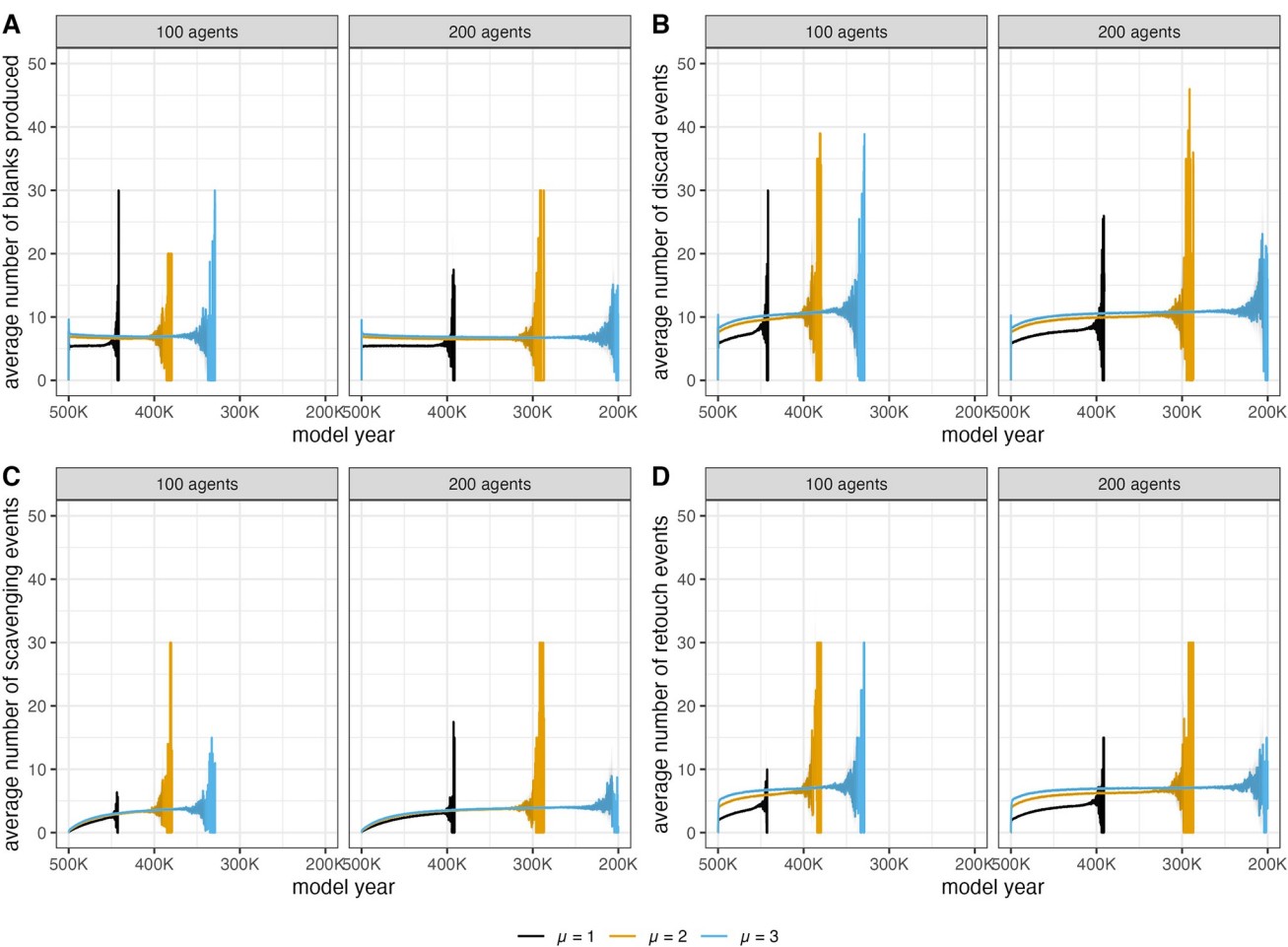

**Fig 8. Comparison of effects of μ and number of agents on number of blanks produced (A), number of objects discarded (B), number of objects scavenged, and number of objects retouched (D) per "model year" (timestep) during model run.** Variation on the right side of the graph produced because some model runs last for more "model years" in order to simulate every agent passing through the landscape.

to only impact the behavioral events when that size preference is strict, resulting in fewer scavenging, retouching, and discard events.

## Grid square level findings

There is a lot of variation in the resulting landscapes between model runs of the same experiment because the path that each agent walks during a model run is unique. To understand more about how this stochasticity affected the outputs, I calculated the coefficients of variation (COV) for each output variable for all 50 model runs performed for each set of experimental parameters. Recycling incidence values vary the most across the landscape with a mean COV of 1.68. All the behavioral events (i.e., discards, scavenging, retouching, grid square encounters) are slightly less variable across the landscape with mean COVs ranging between 1.55 to 0.52. The least variable of these is grid square encounters; this suggests that grid squares are evenly occupied by agents over the course of a model run. Nodule counts vary more than flake counts with mean COVs of 1.36 and 0.83, respectively. Cortex ratios are the least variable across the landscape of all the outputs with a mean COV of 0.09. The variation of model outputs is variously affected by the model parameters. The direction of the effect on variation

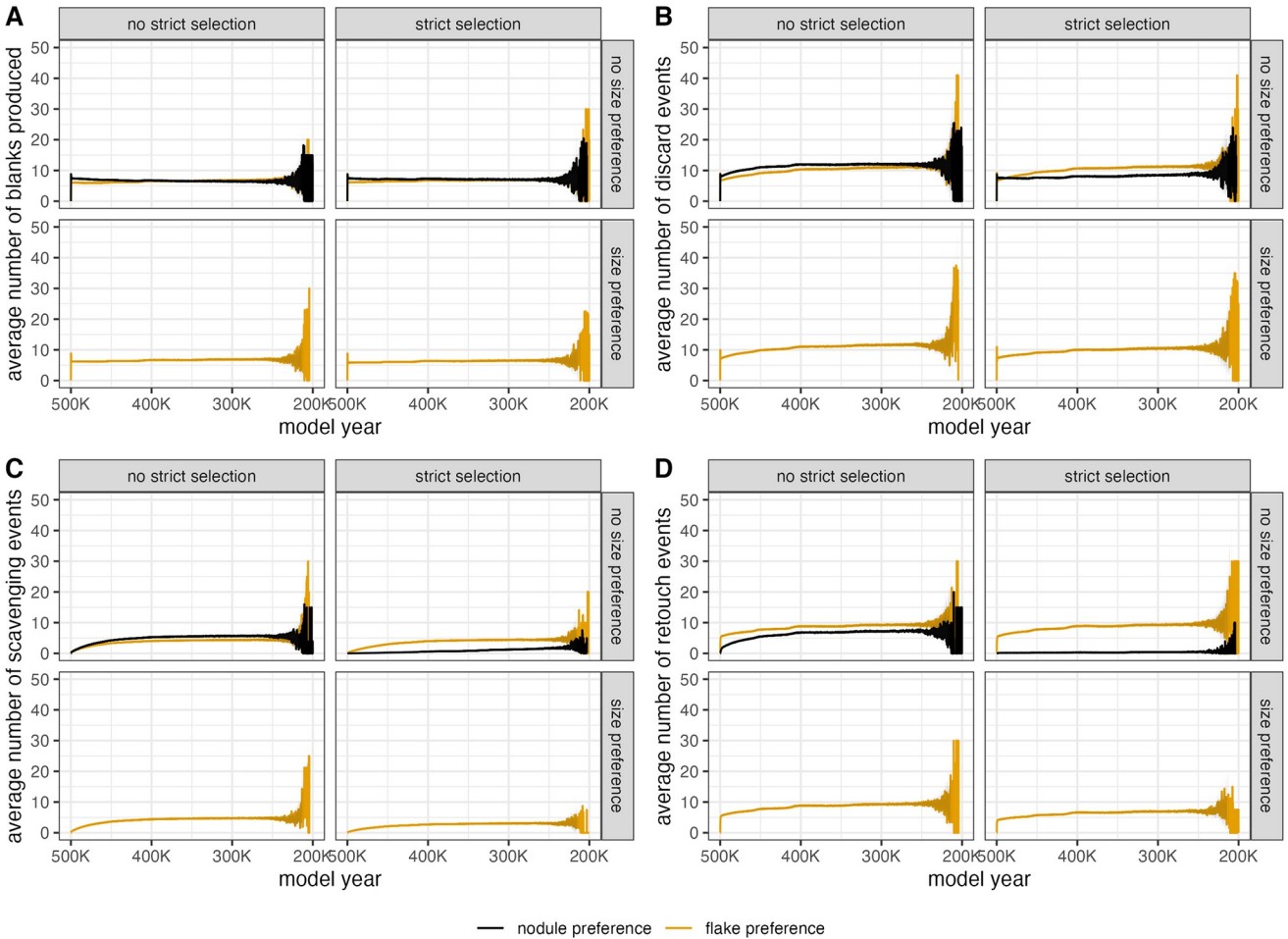

**Fig 9. Comparison of effects of selection parameters on number of blanks produced (A), number of objects discarded (B), number of objects scavenged, and number of objects retouched (D) per "model year" (timestep) during model run.** Variation on the right side of the graph produced because some model runs last for more "model years" in order to simulate every agent passing through the landscape.

caused by each parameter was investigated by performing linear regression for each model parameter on the COVs of each output variable (S7 & S8 Figs). In general, when more agents occupy the landscape and each agent spends more time within the landscape, variation is reduced for all outputs. This is consistent with other findings of decreased variability in assemblages with higher occupation intensity and higher movement redundancy [37, 62]. For selection parameters, the results indicate that limitations on objects that agents are willing to scavenge can cause behavioral events and assemblage characteristics to be less evenly patterned across the landscape (increased COVs). This is likely because in the model agents are somewhat restricted to performing behaviors at locations with suitable scavenging materials. Finally, recycling frequency also impacts variation of model outputs. Increased recycling probability, as determined by high scavenging probability and low blank probability, reduces the variation in recycling incidence values across a landscape, but increases the variation in the occurrence of other behaviors.

**Variation in recycling incidence.** When considering the entire landscape, recycling incidence values increase when recycling behaviors are more frequent. However, recycling

incidence values can also be highly variable across the landscape. This is particularly true when recycling is infrequent and when the landscape is occupied by relatively few agents who spend small amounts of time within the landscape. The question is whether this variation is informative in similar ways that the variation in cortex ratios has been shown to be informative about mobility [37].

To test this, I compared the coefficients of variation of recycling incidence calculated across the landscape for each run of the model across different parameter values via Wilcoxon rank-sum tests. The results show that there is significantly more variation in recycling incidence across the landscape when recycling happens infrequently (scavenging probability is lowest and blank creation probability is highest) (Fig 10). Conversely, all assemblages on the landscape will have relatively similar proportions of recycled artifacts when recycling is frequent.

The reduced variation in recycling incidence with increased recycling frequency holds regardless of occupational intensity (i.e., number of agents) and whether or not every agent

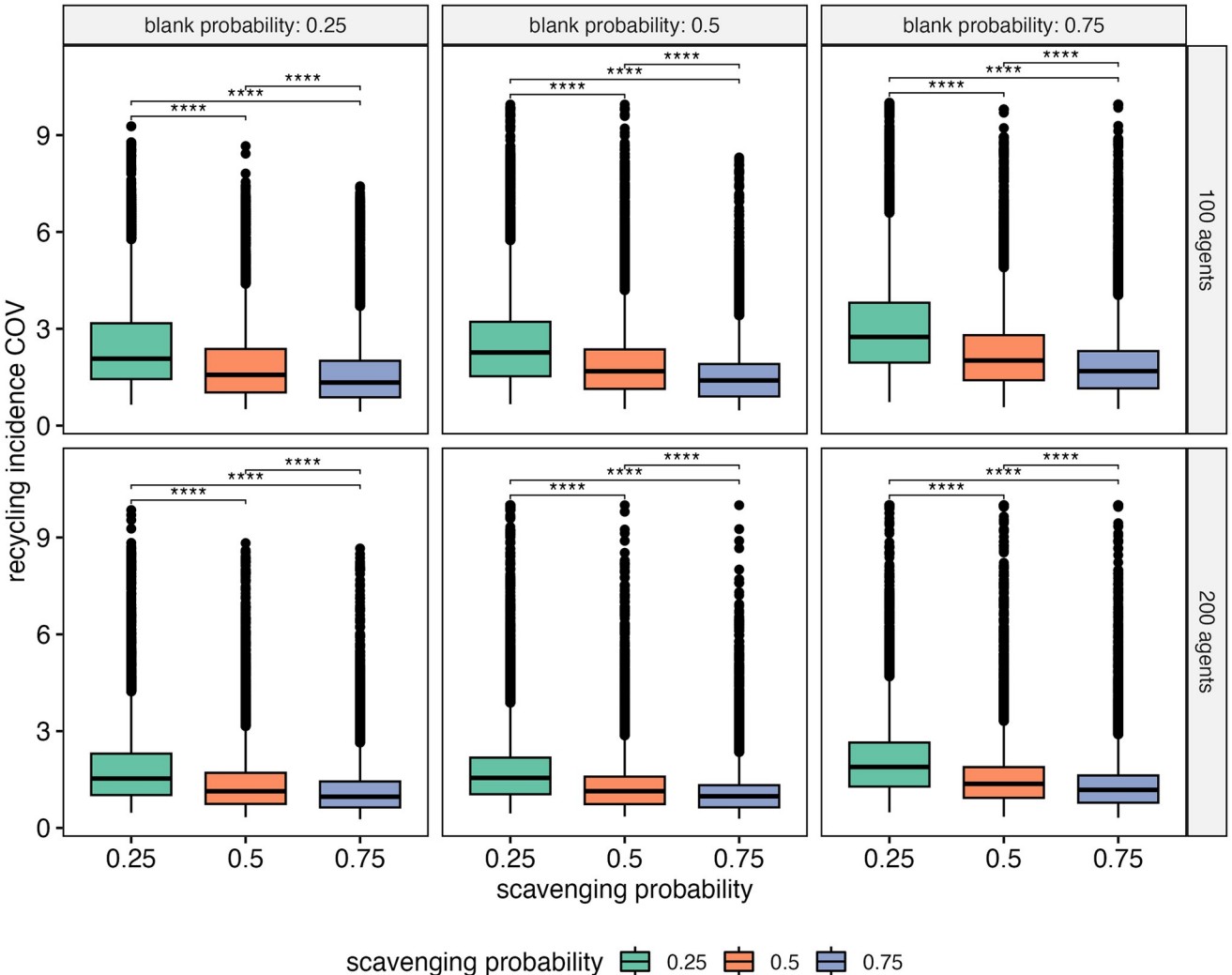

**Fig 10. Boxplots of coefficient of variation (COV) of recycling incidence across all grid squares from each model run.** Wilcoxon rank-sum tests show significant differences in COV between different scavenging probabilities (**** = Bonferroni adjusted p values less than 0.0001) Only data from model runs where agents had one of two technology types (overlap = 1) are used.

has a unique technological tradition. Increased subsequent occupations of the landscape (more agents during model run) serve to further reduce the variation of recycling incidence between assemblages, because more recycled artifacts are created and discarded over the course of archaeological record formation. Additionally, as each agent spends more time on the landscape (increasing value2 of μ), the variation in recycling incidence values decreases (S9 Fig). This is consistent with what Davies and colleagues [37] find for the variation in cortex ratio with increased values of μ.

The variation in recycling incidence increases when agents can carry around more artifacts because more recycled artifacts can be removed from the landscape over the course of a model run (S9 Fig). Selection criteria appear to minimally affect the variation in recycling incidence across the landscape (S10 Fig). A flake preference and a nodule preference when scavenging result in similar amounts of variation, as does strict and non-strict selection. A size preference when scavenging increases the variation in recycling incidence to a greater extent than the other two selection criteria.

In general, any parameters that result in more recycled objects being discarded on the landscape lead to a reduction in the variation in recycling incidence. Interestingly, this pattern does not apply to the other model outputs. High frequency of recycling behaviors increases the variation in flake counts, nodule counts, cortex ratios, number of discard events, and number of retouch events (S7 & S8 Figs).

**Recycling incidence and behavioral events.**   One goal of this study was to understand how recycling incidence relates to other behaviors that occur across the landscape. To this end, I calculate Spearman's rank correlations between recycling incidence and the following model outputs: number of artifacts discarded into a grid square, number of artifacts scavenged from a grid square, number of times an agent occupied a grid square, and number of artifacts retouched at a grid square (Fig 11). Correlations were calculated for each grid square individually across all model runs. The correlations between recycling incidence and behavioral events are on average positive, but some model conditions do produce negative correlations. This suggest that the proportion of recycled artifacts in an assemblage may not always be informative about the behaviors that occurred at a site.

When looking at the effects of different parameter values on these correlations between recycling incidence (RI) and behavioral events, each agent spending more time on the landscape (high μ) causes recycling incidence to become uncorrelated with the number of behavioral events that occur at a grid square (Fig 12), except for the number of retouching events. This makes sense because recycled artifacts are predominantly flakes, which must be retouched by agents to become identifiably recycled objects. In all other cases, agents spending more time on the landscape means they reorganize the discard record over and over again, masking its relationship between behavioral events.

In terms of selection scenarios, correlations between recycling incidence and behavioral events remain relatively unchanged (S11 Fig). This is consistent with the model trend results presented above. The selection parameters result in relatively similar trends in all behavioral outputs (i.e., number of discard events, scavenging events, retouch events, and encounters), so any effect of selection on these correlations would be primarily driven by the effect of selection on recycling incidence.

In terms of frequency of recycling behaviors, increase in scavenging probability reduces the correlations between recycling incidence and behavioral events (S12 Fig). This is consistent with scavenging for the purpose of recycling functioning as a removal process for the use of stone tools elsewhere on a landscape [38], supporting the proposition that recycling is best understood within the context of landscape-level analysis of the archaeological record.

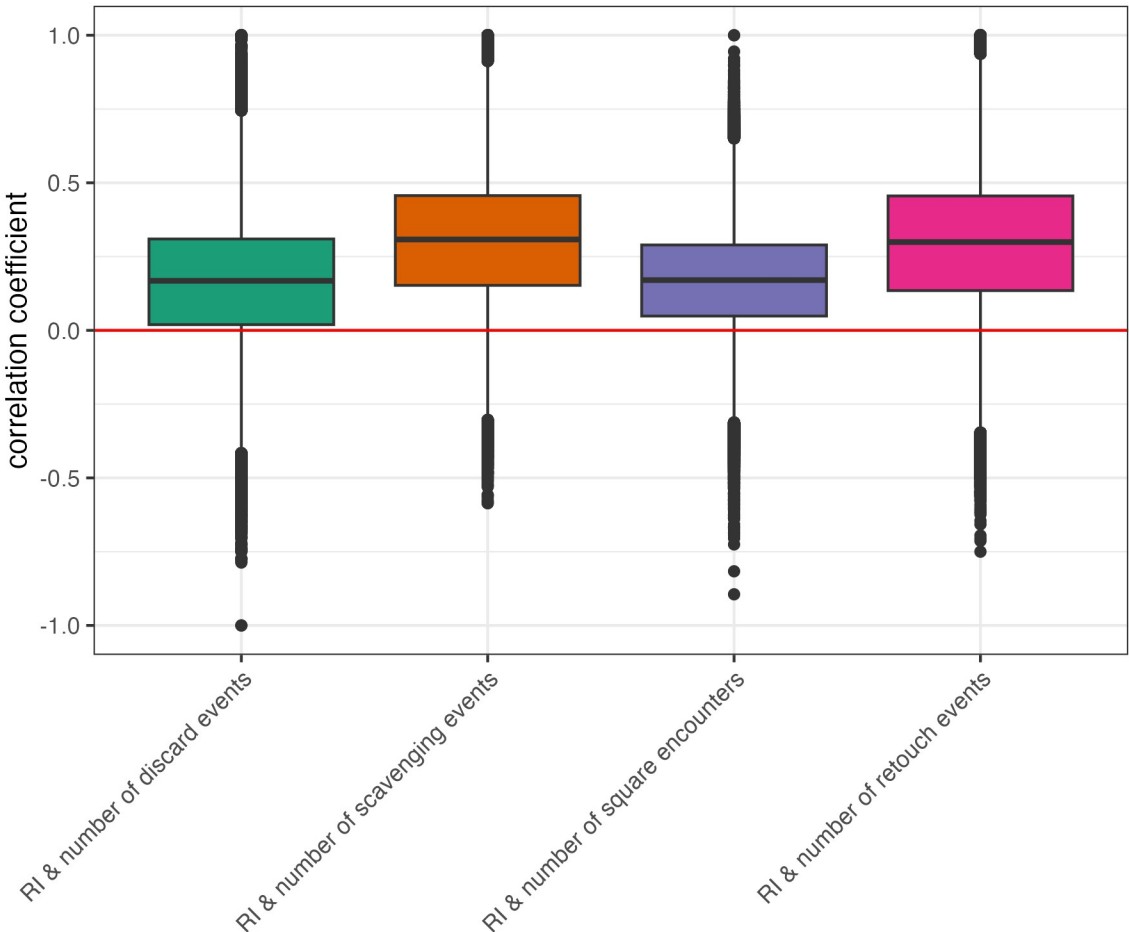

**Fig 11. Average correlations with recycling incidence measured for each grid square across simulation parameters.** From left to right, the correlations are between recycling incidence and: number of discard events, number of scavenging events, number of grid square encounters, and number of retouch events.

**Recycling incidence and assemblage density.** Because recycling incidence is tracked as the proportion of recycled artifacts in an assemblage, it is important to understand if recycling incidence values are determined by assemblage density (as measured by object counts for this model). Assemblage density is a measure of accumulation [62], whereas recycling incidence values are determined by how much things move and how often they are recycled. The strongest positive relationships between recycling incidence and assemblage density (Fig 13) occur when agents are more frequently retouching flakes (low blank creation probability) and exiting the landscape quickly (low μ values). As scavenging probability increases, the relationship between recycling incidence and assemblage density becomes weaker and less positive, especially when agents spend more time in the landscape (high μ).

Increased repeated occupation of the landscape (i.e., more agents) most strongly affects the relationship between recycling incidence and assemblage density when scavenging is frequent (S13 Fig). These conditions result in negative relationships between recycling incidence ratios and assemblage densities, albeit very weak. This weak negative relationship occurs because there is a mismatch in the increases of number of recycled objects compared to that of object count overall when more agents occupy the landscape during a model run and recycling is

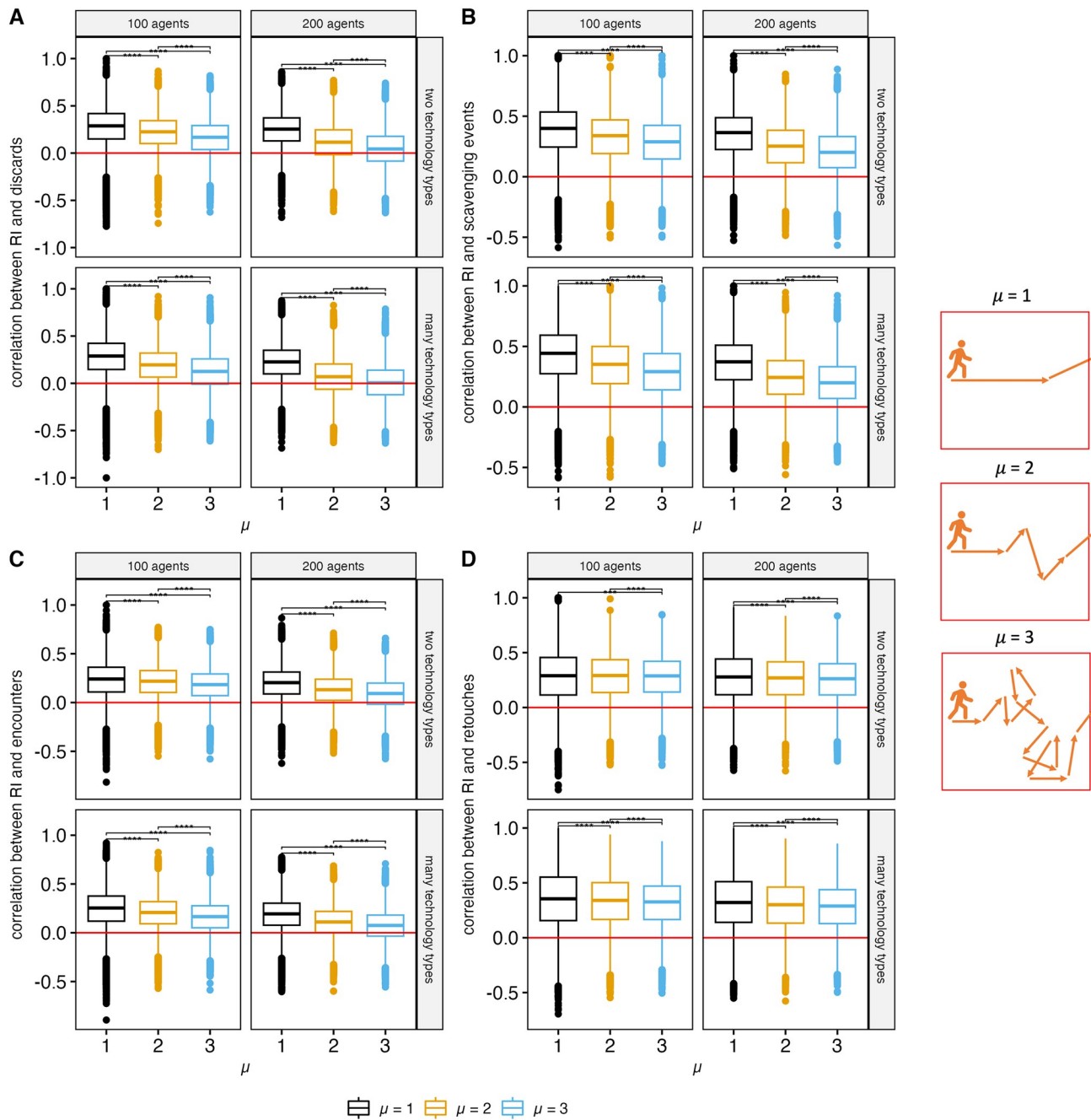

**Fig 12. Correlations with recycling incidence measured for each grid square displayed by number of agents, technological scenario (overlap parameter), and value of μ.** The correlations are between recycling incidence and: number of discard events (A), number of scavenging events (B), number of grid square encounters (C) and number of retouch events (D). Wilcoxon rank-sum tests show significant differences in correlations between different values of μ (**** = Bonferroni adjusted p values less than 0.0001).

frequent. As a result, high object counts drive down recycling incidence values by outpacing the increased number of recycled objects. This means that if a landscape experiences frequent recycling by increasingly large amounts of stone tool-using groups, it is possible that the highest proportions of recycled objects will occur in smaller assemblages.

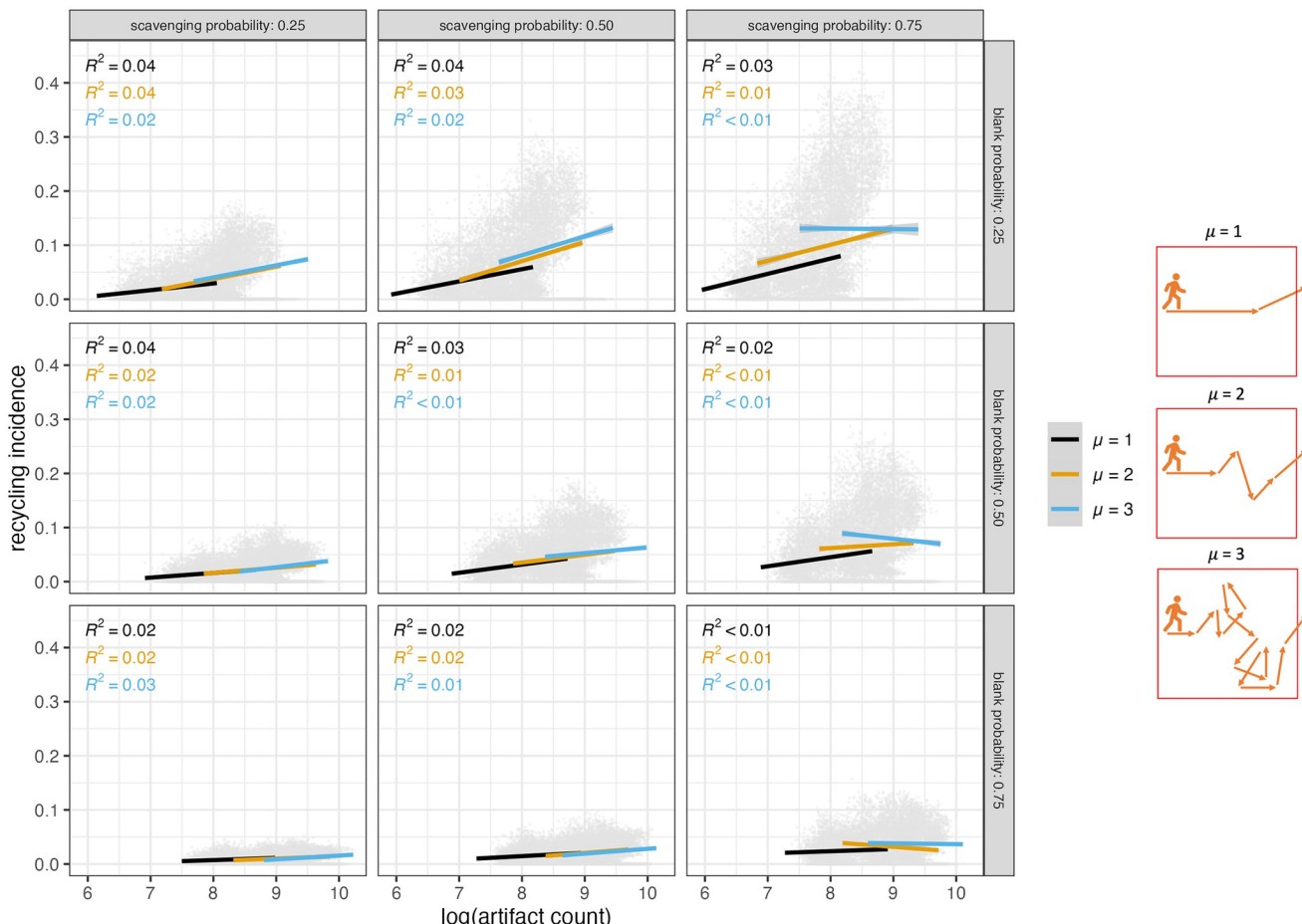

**Fig 13. Relationship between recycling incidence values and logged artifact counts for each grid square.** Results shown for model runs where agents have one of two technology types (overlap is 1) and only 100 agents occupy the landscape during model run. Linear relationship shown for each μ value. Equation and R squared value for each line given in upper left corner of each panel.

The relationship between recycling incidence and object counts only weakly related across all selection preference scenarios (S14 Fig). Instead, the relationship between recycling incidence values and assemblage densities is primarily driven by parameters that allow for increased recycling incidence values and more diversity in assemblage densities. When more objects are being created (high blank probability, more agents, high values of μ), the distribution of assemblage sizes is less skewed (S15 Fig). It is this more even distribution of assemblage sizes that allows for weaker relationships with recycling incidence.

**Recycling incidence and cortex ratios.** Another relationship of interest is that between recycling incidence and cortex ratios within grid squares. Cortex ratios measure artifact movement [37, 58, 62–66], so the relationship between these two metrics can help determine how recycling incidence values depend on the addition and/or removal of artifacts from assemblages.

The cortex ratios produced in this model are consistent with findings by Davies and colleagues [37]. Cortex ratios are lower than 1 when agents exit the landscape quickly (low μ); Cortex ratio values approach 1 and become less variable as agents spend more time on the landscape (high μ). Lower μ values also foster more fragmentation of the discard record [41], because agents have less time on the landscape to discard objects (see discard trends in Fig 7).

The values of cortex ratios produced during model run are largely determined by whether agents have a flake preference or a nodule preference. In this model, flakes are simulated as completely cortical objects, meaning that when a flake is removed from a nodule, the remaining nodule becomes increasingly non-cortical. As a result, a nodule preference fosters the removal of non-cortical elements from assemblages, resulting in cortex ratios above 1 (S16 Fig). Conversely, when agents prefer flakes, cortex ratios are below 1 because agents preferentially remove completely cortical objects from the landscape. Because of this, the relationship between recycling incidence values and cortex ratios of assemblage must be investigated for flake and nodule preferences separately.

When agents have a nodule preference, recycling incidence is only very weakly related to cortex ratio in any given assemblage. When agents have a flake preference, the relationship between recycling incidence and cortex ratios is stronger and more interesting. There are strong negative relationships between recycling incidence and cortex ratios when cortex ratios are tightly clustered around a value of 1 (Fig 14). A negative relationship between recycling incidence and cortex ratio means that assemblages experiencing relatively little artifact dispersal are also the assemblages with lower proportions of recycled artifacts. This relationship is strongest when agents spend more time on the landscape (higher values of μ) (S16 Fig). This means that when there more opportunities for local discard of objects, it is more likely that

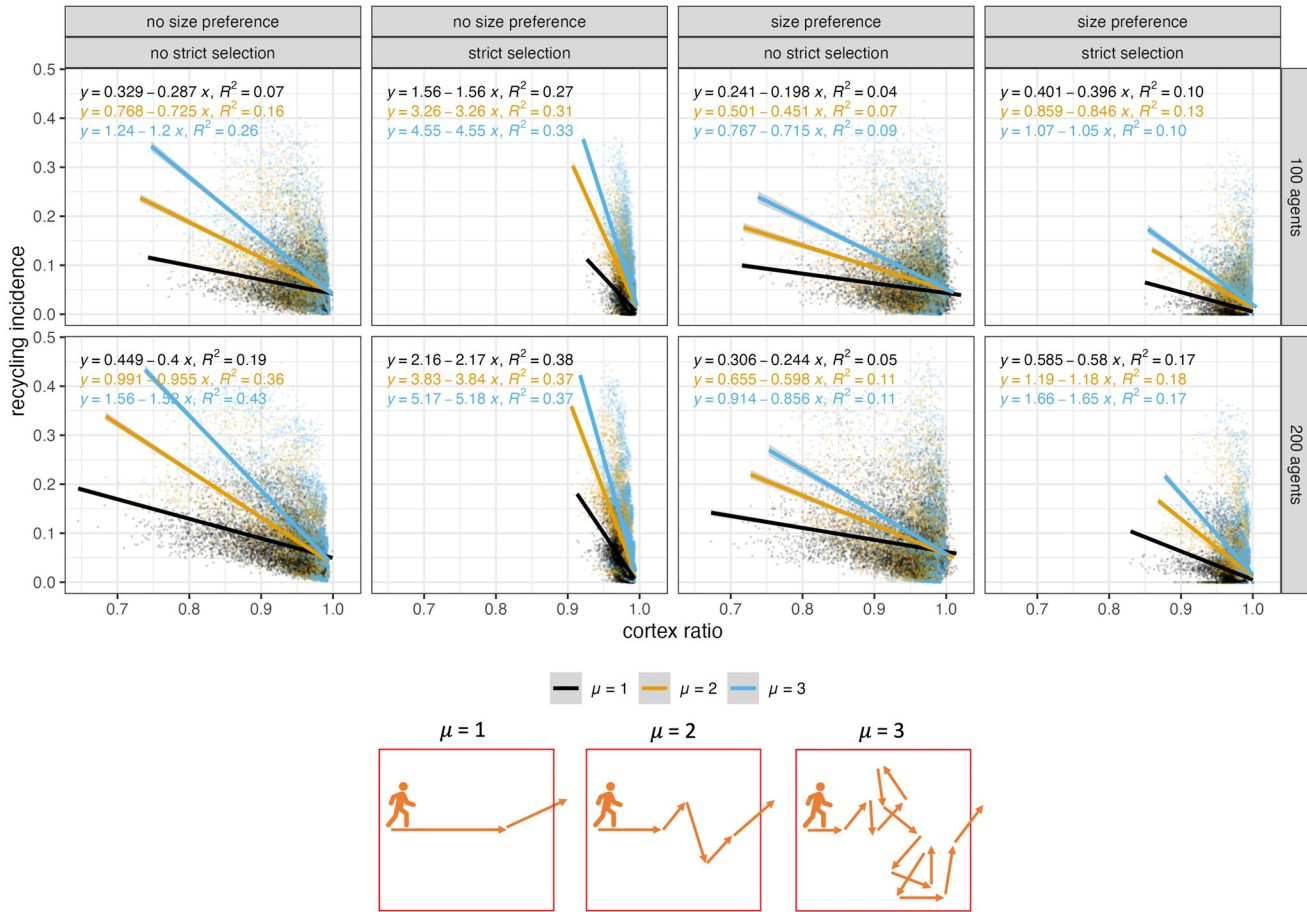

**Fig 14. Relationship between recycling incidence values and cortex ratios across all assemblages for each model run.** Data displayed for model runs where agents have one of two technology types and a flake preference.

recycling incidence will be highest in those assemblages where the most objects have been removed.

It is important to consider that cortex ratios and recycling incidence will both be affected by assemblage size. Part of the reason that high recycling incidence values occur in assemblages with more removal of objects is because those assemblages will have fewer objects and therefore recycled artifacts can make up greater proportions. However, when looking at the relationship between cortex ratios and assemblage density, the positive correlation between these two metrics is not extremely strong (S17 Fig). This means that even when controlling for the effect of assemblage density on recycling incidence, there is still a significant negative relationship between recycling incidence and cortex ratios.

**Hotspots of behavior.** Understanding the spatial structure of recycling behaviors in the archaeological record is important. This was accomplished by examining how concentrations of high recycling incidence values overlapped with high concentrations of other behaviors and objects. Recycling incidence values and the other layer outputs at each grid square were compared to their queen neighbors to calculate local G statistics via the spdep package in R [67]. Grid squares with the highest local G values (more than two standard deviations from the mean) were compared to determine how many of these high-value squares overlapped with the grid squares with the highest local G values for recycling incidence.

There is relatively little overlap of high concentrations of recycling incidence (RI) and high concentrations of other outputs (Fig 15), despite the fact that on average 5% of the landscape has high value hotspots for each of the outputs individually (S18 Fig). This means that areas of

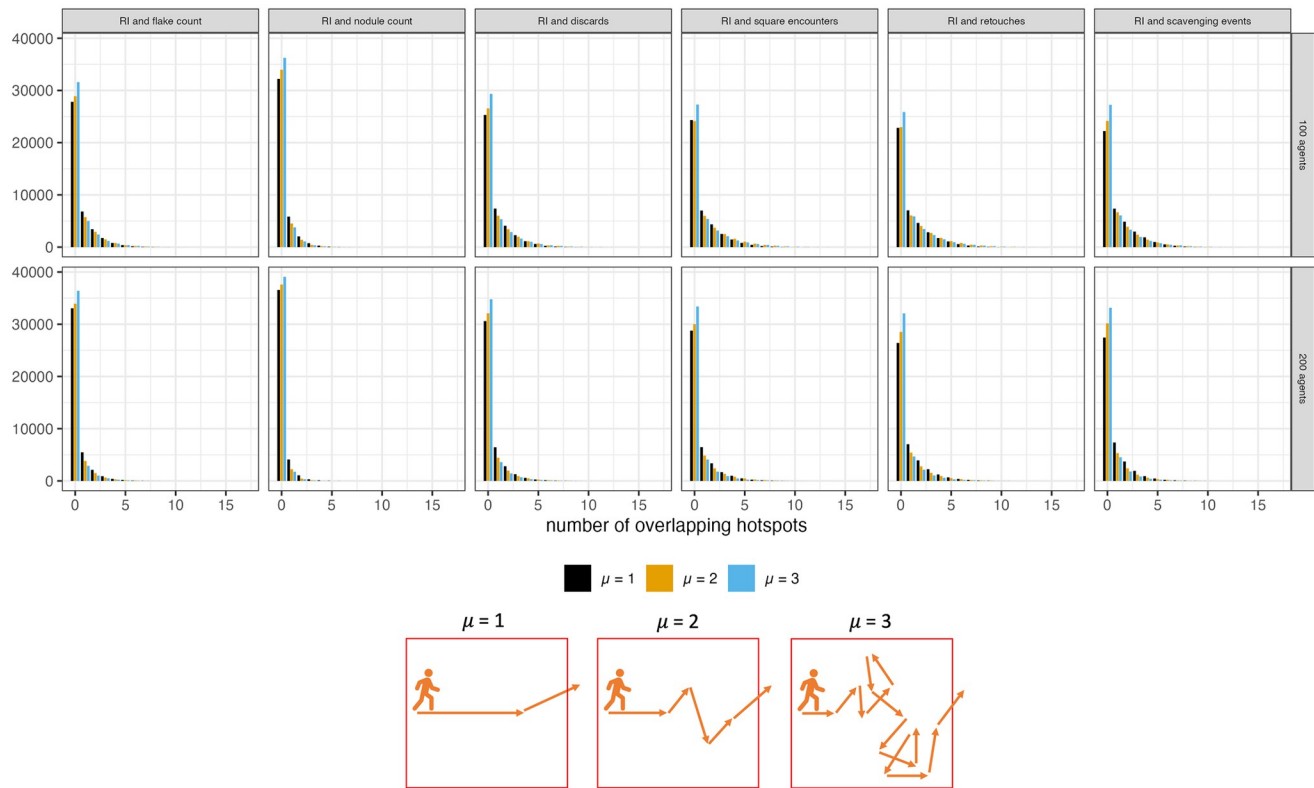

**Fig 15. Distributions of hotspot overlap counts for overlaps with high recycling incidence (RI) across all model runs.** Result here show the counts for the two technology scenarios (overlap is 1) only.

high recycling incidence occur in different locations on the landscape than the hotspots for the other outputs.

To investigate which parameters result in no overlap of hotspots and which parameters increase the amount of hotspot overlap I used zero-inflated regressions via the pscl package in R [68]. Zero-inflated regressions are used to model count data that have an excess of zero counts, such as is the case for the number of overlapping hotspots. These regressions accomplish this by combining two models: a Poisson count model and a logit model for the excess zeros.

Parameters that cause lower recycling incidence values across the landscape, such as a size preference, strict selection, larger minimum flake size for selection, increase the likelihood that there will be no overlap with hotspots of recycling incidence (Fig 16). Interestingly, parameters that result in more objects being created and discarded (high μ, more agents, high blank probability) also appear to increase the likelihood of no overlap with hotspots of recycling incidence (Fig 16). This suggests that high recycling incidence is a poor indicator of where agents perform other behaviors in a denser archaeological record.

For the most part, parameters that result in no overlap of hotspots also cause in smaller counts of overlapping hotspots and vice versa (Fig 17). Interestingly, strict selection has a large positive effect on the number of overlapping hotspots despite also resulting in a lack of overlap between hotspots of recycling incidence and flake counts and that of recycling incidence and grid square encounters. This could be because, strict selection forces activities on the landscape

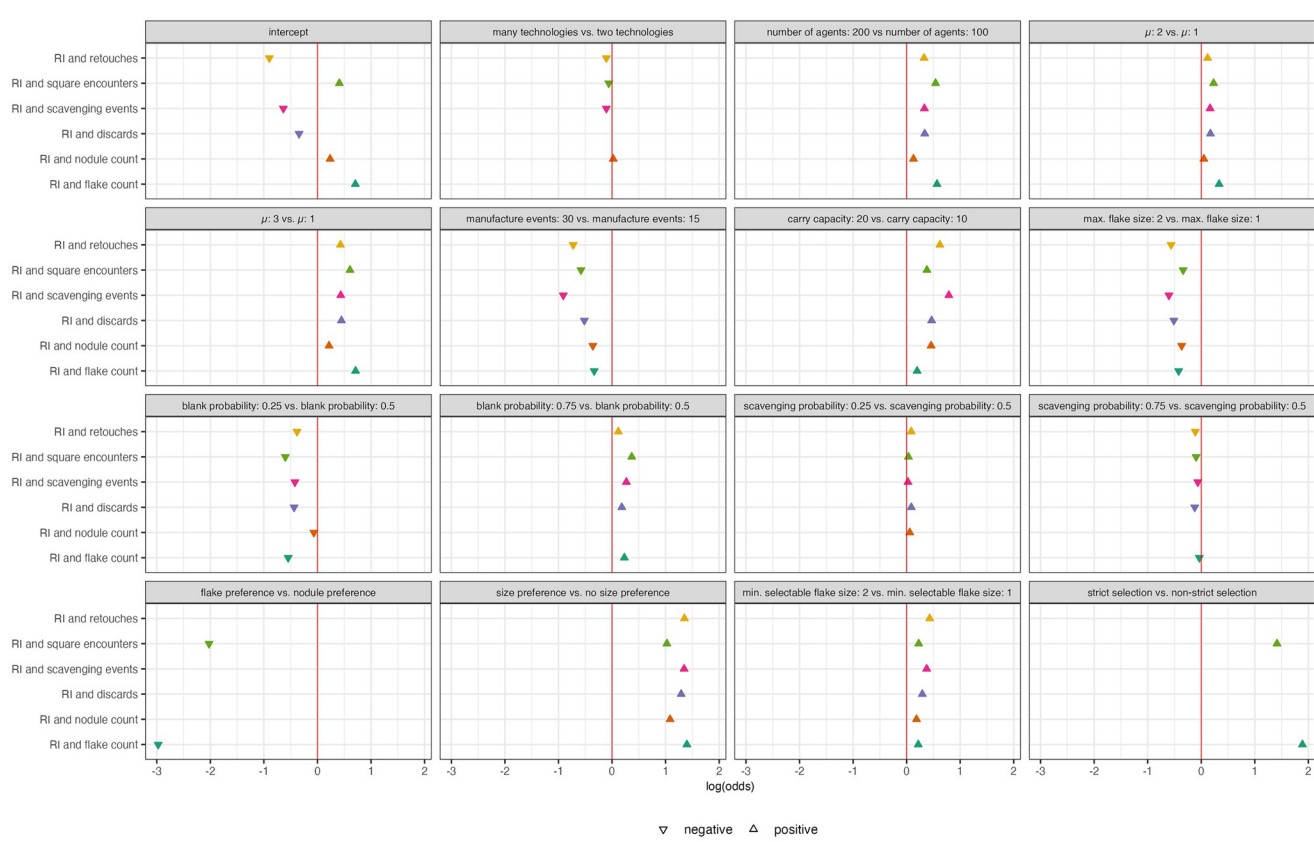

**Fig 16. Zero-inflated log odds regression coefficients for effect of model parameters on counts of hotspot overlap.** Results are displayed for excess of zero counts for overlap of hotspot squares of recycling incidence(RI) and other output variables. Only significant coefficient values are plotted.

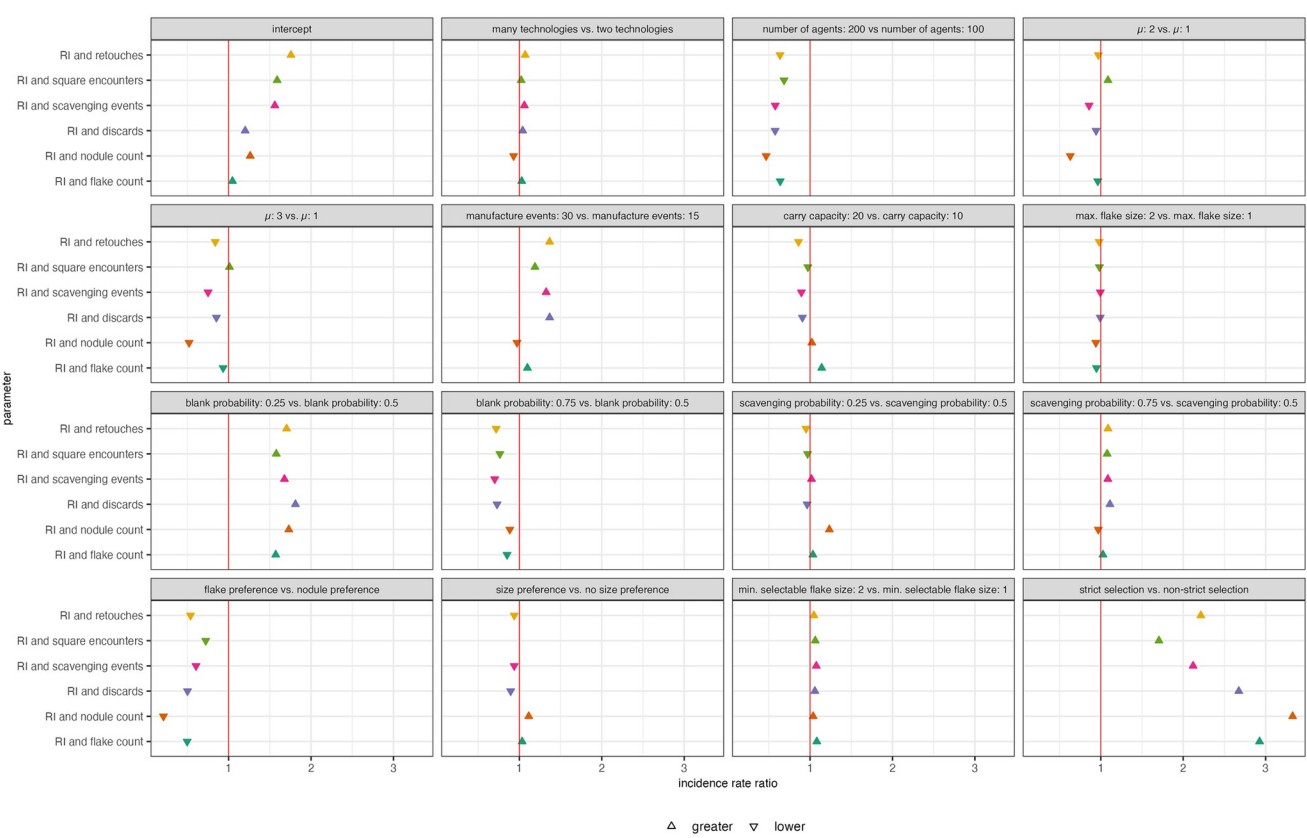

**Fig 17. Poisson incidence ratio regression coefficients for model parameters.** Results are displayed for counts for overlap of hotspot squares of recycling incidence (RI) and other output variables. Only significant coefficient values are plotted.

to happen in the same area by limiting the locations from which agents can scavenge artifacts. For example, if the agent does not scavenge artifacts at one location due to the limitations of its selection criteria, then the agent is also less likely to perform other behaviors there because it will not have any objects to manufacture or discard.

The results of the hotspot analyses indicate that high recycling incidence values rarely occur in the same location as high object counts, which is akin to assemblage density in the archaeological record. This supports the finding of a weak relationship between recycling incidence values and assemblage density discussed above.

Given the effects of recycling behaviors on the patterning produced in cortex ratios, I tested whether another archaeological proxy was affected. Specifically, I looked at whether the proportion of retouched tools in an assemblage was positively correlated with occupation intensity or occupation redundancy as archaeologists have proposed [69–72]. This was accomplished by looking at hotspot overlap between grid squares with high encounter rates and grid squares with high proportions of retouched artifacts. In the context of this model, retouched artifacts are those that have been "knapped" following their initial removal from a nodule. There is an excess of zero counts for overlap in hotspots of retouched artifact proportions and grid square encounter rates in this model (Fig 18A). This is not because these two metrics are uncorrelated within assemblages. In fact, proportion of retouched artifacts is predominantly positively correlated with number of grid square encounters (S19 Fig). The lack of hotspot overlap simply means the highest values of each do not necessarily occur at the same location on the

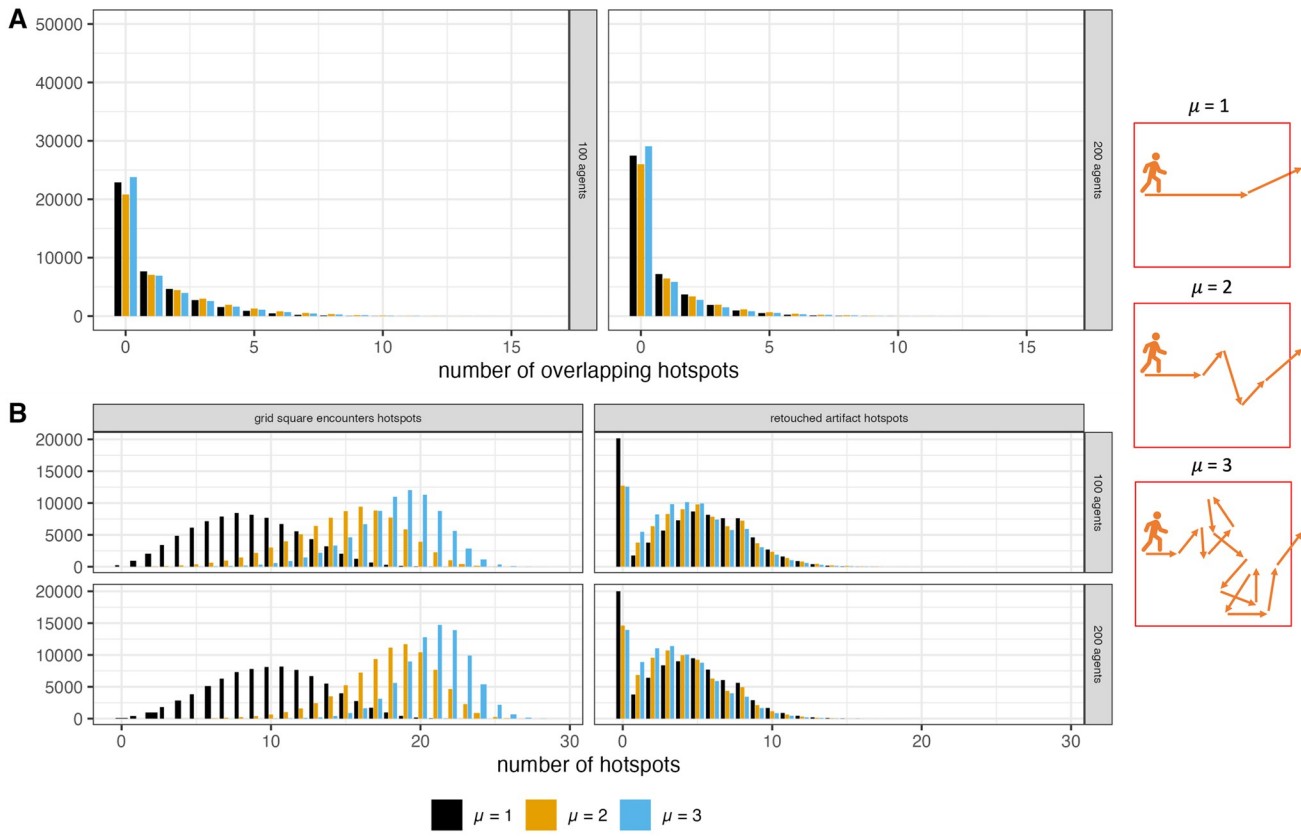

**Fig 18. Hotspot overlap for retouched artifact proportions and grid square encounters (A) and distribution of hotspots for each (B).** Hotspot and overlap counts shown for all model runs.

landscape. This is likely primarily driven by a lack of hotspots for retouched artifact proportions (Fig 18B).

Agent behaviors hotspots tend to cluster in the center of the modeled landscape, especially when each agent spends more time on the landscape (high μ). Conversely, recycling incidence hotspots do not cluster centrally. The combination of these two results means that overlap of recycling incidence hotspots and hotspots of agent behavior typically occur in the center of the landscape, but rarely with more than 2 grid squares defined as hotspots overlaps per model run. The results of the hotspot overlap analyses appear to suggest that high values of recycling incidence are a relatively poor proxy for the location of frequent behavioral events on a landscape.

## Discussion

A major question this paper sought to answer is whether recycling incidence, or proportion of recycled objects found in an assemblage, is indicative of how much recycling occurred. The results demonstrate that recycling incidence values increase on a landscape scale when there is a higher probability of recycling behaviors. This increase in recycling incidence is accompanied by overall less variation in the proportions of recycled objects per assemblage across the landscape when recycling behaviors are frequent, however variation still exists. This is important because it means that different spots on the landscape will provide different insights into recycling frequency, potentially masking a different pattern on a wider scale. As a result, this

model demonstrates that a landscape-scale approach is necessary for understanding recycling behaviors, particularly in cases of frequent recycling behaviors.

The variation in recycling incidence between assemblages decreases over time (S20 Fig), meaning that assemblages which have been accumulating for longer are more likely to have similar signals of recycling incidence. This is also true when each agent spends an extended amount of time on the landscape (high μ) and more agents occupy the landscape. This reduced variation of recycling incidence with increased μ mirrors the patterns of reduced variation found by Davies and colleagues [37] for cortex ratios and by Barrett [41] for distribution of reduction set sizes. When found archaeologically, this lack of variation can demonstrate long term regularity in landscape use [41, 73]. In the context of recycling behaviors, a lack of variation in proportions of recycled artifacts across a landscape indicate long-term stability in frequency of recycling behaviors.

Another important result from this model is the confirmation that, in pure surface contexts, recycled artifacts will typically have been exposed for longer compared to non-recycled artifacts. Furthermore, recycled artifacts will occur more frequently in older assemblages. These findings fit well with Camilli and Ebert's [35] hypothesis that exposure of artifacts facilitates artifact discovery and therefore scavenging and subsequent recycling. Although this model focuses on surface contexts, the established relationship between exposure and recycling sets up a further hypothesis: when comparing any type of deposits, those that have been exposed for longer will have larger amounts of recycled objects. This hypothesis can be tested by comparing the numbers of recycled artifacts in archaeological assemblages with different accumulation intervals.

### Location of archaeological signatures of recycling

An important result of this model is the finding that areas of high recycling incidence are not necessarily also places where lots of other occupational activities happened. For example, the overlap between areas of high recycling incidence and those with high numbers of scavenging events was less than 1% of landscape on average, even though 9% of the landscape is identified as a hotspot of scavenging activity on average. Archaeologists often use proportions of recycled artifacts to discuss the importance or intensity of recycling behaviors at a particular site. However, the results of this model demonstrate that in some contexts there will be a disconnect between where recycled artifacts are found and where occupation or scavenging behaviors happened.

Although the hotspots do not overlap, recycling incidence is consistently positively correlated with counts of behavioral events. That being said, increased recycling behaviors *reduce* the strength of this correlation, making values of recycling incidence increasingly less informative about the frequency of behavioral events that occurred at a particular site when recycling is frequent. To make matters more complicated, these assemblage-level correlations are influenced by mobility. When each agent spends more time occupying a landscape, the proportion of recycled objects at a location is comparatively less reliable for interpreting discard, scavenging, occupation, or retouch behaviors. In this model, mobility patterns are determined by value of μ, which modulates the relative frequencies of long-distance and short-distance steps in a Lévy walk. Long-distance steps are more likely at low values of μ, causing the agents to exit the landscape more quickly. High values of μ result in shorter step lengths, causing agents to spend more time in the landscape and have more redundant movement paths. In a logistical mobility system, low μ values would represent landscape areas where logistical forays occur, whereas high μ values simulate movement that would be more characteristic of areas around a base camp [58]. This means that if recycling is occurring within a logistical mobility system,

archaeologist should expect higher recycling incidence values to be more informative about the location of other behaviors in landscape areas where logistical forays occur. Conversely, if recycling behaviors are occurring in base camp settings, recycling incidence values will be less variable across the landscape and therefore less informative about where foragers are performing other stone tool use behaviors.

In a residential mobility scheme, place use is more equal across a landscape, so landscapes characterized by different frequencies of long-distance and short-distance movements will be dictated by how often and how far groups decide to move their residences. Studies of ethnographic hunter-gatherer populations have demonstrated that these decisions are primarily determined by factors such as habitat quality, subsistence strategy, group sizes, and population density [74, 75]. This means that when interpreting the appearance of recycling behaviors for populations practicing residential mobility, other archaeological data will first need to be incorporated to assess the frequency of long- versus short-distance moves within a landscape.

Although proportions of recycled objects are often not informative about location of behaviors, high assemblage densities will indicate where other agent behaviors occur more frequently in the context of recycling. Object counts are always positively correlated with discard events, and typically very highly positively correlated with scavenging events and grid square encounters (S21 Fig). These relationships are consistent with artifact density being used as an indicator of occupational redundancy, particularly when foragers are drawn to previously deposited materials for the purpose of recycling [36].

Interestingly, the relationship between recycling incidence and other aspects of the archaeological record are only minorly effected by selection preferences during scavenging. The largest impacts are notable when selection is strict, but that is not a typical scenario archaeologically [25]. Yet, in many cases of recycling known from the archaeological record, there does appear to be some sort of size preference when recycling occurs [18, 76–78]. For this reason, it makes the most sense to consider non-strict selection scenarios to establish archaeologically testable patterns. size preference act as an averaging force on assemblage density distributions and concentrations of agent behaviors across the landscape by limiting the locations from which agents are willing to scavenge artifacts. Conversely, size preference increases the variation in recycling incidence across a landscape (S10 Fig). This suggests that a mismatch in the variability of recycled artifact proportions and other indicators of human behaviors across the landscape might indicate some sort of limitation imposed on scavenging behaviors, be that a size preference or any other type of selection criterion.

The results of the model demonstrate that there is a lot of equifinality in the effects of recycling frequency, mobility strategy, and selection criteria on the appearance of recycled artifacts in the archaeological record. For example, both high recycling frequency and a high frequency of short moves by agents (high μ) reduce the variation in recycling incidence values across the landscape. As a result, interpreting an archaeological record affected by recycling behaviors will require considering many different aspects of the record at once. In Table 1, I outline a few conditions that co-occur under distinct recycling regimes.

## Recycling and archaeological proxies

Another driving question of creating this model was whether recycling behaviors significantly change other archaeological proxies. For example, the number of retouched artifacts is frequently used to assess occupation frequency of a site [69–71, 79]. Although the model does not differentiate between primary and secondary flaking behaviors, the results from this simulation indicate that artifacts that have been "knapped" at least once following their removal from a core occur in proportions that are correlated with how frequently an assemblage has been

**Table 1. Expected archaeological record conditions for various recycling behavior frequency, mobility styles, and selection criteria.**

| Recycling frequency | Mobility style: Frequent, short moves (high μ) | Mobility style: Infrequent, long moves (low μ) | Selection behaviors: Flakes preferentially scavenged | Selection behaviors: Flakes of certain size preferentially scavenged |
|---|---|---|---|---|
| High | • Highest recycling incidence (RI) values<br>• Less variation in RI between sites<br>• RI uncorrelated with assemblage density<br>• RI less correlated with location of behavioral events<br>• RI weakly correlated with cortex ratios | • High recycling incidence (RI) values<br>• Slightly elevated variation in RI between sites<br>• RI weakly positively correlated with assemblage density<br>• Some possible concentrations of high RI values | • High recycling incidence (RI) values<br>• Less variation in RI between sites<br>• RI weakly correlated with cortex ratios | • High recycling incidence (RI) values, but lower relative to no size preference<br>• Slightly elevated variation in RI between sites<br>• RI more positively correlated with location of behavioral events |
| Low<br>(RI values relatively unaffected by mobility and selection) | • Low recycling incidence (RI) values<br>• Lower variation in RI between sites<br>• RI weakly positively correlated with assemblage density<br>• RI less correlated with location of behavioral events<br>• RI strongly negatively correlated with cortex ratios | • Low recycling incidence (RI) values<br>• More variation in RI between sites<br>• Some possible concentrations of high RI values<br>• RI slightly positively correlated with location of behavioral events | • Low recycling incidence (RI) values<br>• More variation in RI between sites<br>• High RI values where cortex ratios closer to 1 | • Low recycling incidence (RI) values<br>• Elevated variation in RI between sites |

visited by agents. This is true at all levels of recycling behaviors (S21 Fig), although the correlation is lower when recycling is more frequent.

Cortex ratios are another archaeological proxy, which been shown to be informative about mobility [37, 58, 62–66]. The cortex ratios produced across all model runs are consistent with those modeled by Davies and colleagues, where more time spent in the landscape by each agent (high μ) reduces the variability of cortex ratios across a landscape [37]. However, when this is broken down by frequency of recycling behaviors, a different pattern emerges. Frequent recycling make cortex ratios more variable independent of the effects of mobility. In fact, at high frequencies of recycling, the effect of μ on cortex ratio variation disappears (S22 Fig). Furthermore, at the highest recycling frequencies, there is actually an increase in variation of cortex ratios across the landscape as agents spend more time in the landscape. This means that cortex ratios be used to understand artifact movement on a landscape only at low levels of recycling; at high levels of recycling this may not be the case. As such, the model shows that cortex ratios need to be interpreted in the context of recycling incidence. If proportions of recycled artifacts are high and less variable across a landscape, indicating high frequency of recycling, then cortex ratios may not behave in the ways described by Davies and colleagues [37].

The relationship between recycling incidence and cortex ratios can be informative. The model demonstrates that high proportions of recycled artifacts are less likely to be in locations where *in situ* scavenging, tool use, and discard occur when artifacts are frequently removed from a landscape. Instead, recycled objects are more likely to be moved around the landscape, ending up in assemblages characterized by frequent artifact movement (i.e., have cortex ratios less than 1).

## Future directions

As it is currently written, this model is not suited for addressing the effect of recycling on some additional aspects of assemblage variation. For example, this model uses a simple core and flake conceptualization of stone tools, ignoring tool types that are often identified in the archaeological record. Tool types are often used to assess the composition of assemblages and relate it to concepts such as intensity of occupation, reduction trajectories, assemblage accumulation intervals, discard rates, and tool use life [80–82]. Many assessment of assemblage composition show it is size-dependent, a property supposed to emerge due to an interaction between occupation span and tool use-life assuming regular discard of tools at a site [82, 83]. However, these proposed relationships do not consider the potential removal of items after initial discard; it would be interesting to see if scavenging affects the size-dependency of assemblage composition especially considering the relationship between recycling and exposure (which is akin to accumulation interval in this model).

Another thing this model does not explore is the effect of artifact use life on the relationship between artifact age and likelihood of it being recycled. In the model, all artifacts have an unlimited use life, meaning they can be knapped again and again until the end of model run. Obviously, in the archaeological record this is not the case; knapping of stone tools is a reductive process, so objects will necessarily have a limited use-life after which they are no longer useable [84]. Other modeling that incorporates use-life of lithic artifacts as a parameter has demonstrated that this parameter can affect maximum transport distances of stone tools and how well that reflects maximum group range [85]. Simulations by Barton and Riel-Salvatore [86] demonstrated that when artifact use-life is expended more quickly, there is an increased proportion of retouched artifacts in assemblages; furthermore, when artifact use-life is expended less quickly, there is more variation in assemblage composition in terms of remaining artifact use-life [86]. More formal modeling by Surovell demonstrates how increased artifact use-life results in more constant ratios of local to transported materials over time [87]. For the model presented here, it is possible that older artifacts, although exposed for longer and therefore more discoverable throughout time, would also have comparatively little remaining utility, forcing groups to scavenge younger artifacts that have more utility. The use of artifacts that have been used the least is a tendency that is documented ethnographically [88]. Stone tool users also do appear to prefer larger artifacts when scavenging for reuse [e.g. 18], which may have something to do with the remaining utility. Future iterations of recycling models could explore how limiting the use-life of artifacts affects the relationship between when an artifact is discarded and how likely it is for that artifact to be recycled. It is likely that artifact use-life would act to create some assemblages without a correlation between exposure time and recycling, much in the way the strict selection scenarios modeled here produced assemblages where recycled artifacts were not those discarded earlier in time.

Simulations by Davies and colleagues have demonstrated the relationship between cortex ratios and assemblage densities is dependent on the consistency of landscape use [62]. They model an "accumulation" scenario where all agent movement follows consistent patterns (constant value of μ for each agent), but the number of agents that occupies the landscape during model run is random. They also model an "occupation" scenario where each agent's movement is dictated by a random value of μ, but the number of agents is held constant. Cortex ratios are only assemblage density-dependent under the "occupation" scenario when landscape use is varied. In the model presented in this paper, with both the number of agents and μ are held constant throughout each model run. This limits any comparison between recycling incidence values and assemblage densities for understanding consistency in landscape use. Further experiments with the model presented in this paper could be done to more thoroughly

investigate the differences in recycling incidence values between an "accumulation" and an "occupation" model *sensu* Davies et al. [62].

## Conclusions

The findings from this model demonstrate that a landscape-level approach is crucial to understanding nuances in recycling behaviors, especially within the context of differing mobility patterns and scavenging preferences. This is true not only for contextualizing proportions of recycled artifacts, but also for understanding how those proportions should be related to other archaeological phenomenon, such as assemblage densities and cortex ratios. As the importance of recycling as a part of stone tool use continues to be recognized at individual sites, archaeologists must now turn to wider regional approaches for understanding how Paleolithic stone tool users employed recycling as part of their technological strategy.

The model presented here is a simplified simulation of recycling behaviors. The resulting patterns of this model, therefore, do not necessarily represent true patterns produced by recycling behaviors in the past, but instead can be used for hypothesis testing and bounding our expectations of archaeological data under different parameters [89–91]. The results of the recycling model demonstrate that archaeological record formation due to recycling behaviors is indeed influenced by differences in mobility strategies and selection criteria of groups. The model also demonstrates that there is spatial structure to the patterns created by recycling behaviors across a landscape. Both of these results highlight the need to consider recycling behaviors within the context of regional landscape use by mobile populations [47].

The model demonstrates that while the number of recycled objects can be indicative of the frequency of recycling behaviors on a landscape, the location of recycled objects does not necessarily inform us about where scavenging behaviors or retouching of scavenged materials occurred [50]. This is particularly true when scavenging behaviors are occurring frequently. The model also demonstrates that decreased variation in proportions of recycled artifacts in assemblages across a landscape is indicative of long-term occurrence of frequent recycling behaviors. Furthermore, this long-term consistency in recycling behaviors in a landscape is likely to result in higher proportions of recycled objects occurring in assemblages that have been accumulating for longer.

Other archaeological metrics used to interpret artifact movement (cortex ratios) and site occupation frequency (retouched artifact counts) still maintain their previously established patterns at low levels of recycling behaviors. At higher levels of recycling behaviors, it is prudent to consider archaeological proxies in the context of recycling. For example, the results of this model demonstrated that the relationship between variation in cortex ratios and movement redundancy is not straightforward when recycling is highly frequent.

It is important to note that all results presented here are more likely to apply in contexts where a layer has been exposed for an extended period, such as surface sites. Previous research demonstrates that the frequency and type of geological events can significantly impact the technological characteristics of scavenged and recycled assemblages [23]. Therefore, it is very likely that including geological events in this model would change the patterns created during archaeological site formation in the context of continuous recycling behaviors across time. This, however, does not take away from the fact that the recycling of stone tools is a powerful force repeatedly rewriting archaeological patterns throughout history. Archaeologists are continually finding new evidence that recycling behaviors were common during the Paleolithic. Despite this understanding, the implications of this behavior have not been thoroughly explored. This model offers insight into how recycling behaviors under different mobility scenarios and with different selection criteria can have large effects on the formation of the

archaeological record over time. In doing so, the findings presented in this paper bring us closer to being able to interpret the stone tool record most accurately.

## Supporting information

**S1 Text. Overview, design concepts, and details for extended recycling model.** Description of the recycling agent-based model used in this paper following the ODD protocol. (DOCX)

**S1 Fig. Relationship between recycling intensity (proportion of recycled artifacts in a grid square) and skewness of distribution of each artifact's year of initial discard for each grid square.** Results show for model runs when agents have one of two technology types (overlap = 1) and only 100 agents occupy the landscape during model run. Negative skew indicates artifacts in the assemblage are first discarded earlier in model run. Positive skew indicates artifacts are discarded later in model run. Linear relationship shown by dark blue line. R squared values given in upper left corner of each panel. (TIF)

**S2 Fig. Total counts of recycled objects (A) and average counts of recycled objects per model run (B).** Results shown for model runs when agents have one of two technology types (overlap is 1). Error bars in panel B represent the 95% confidence intervals on the average per model run. (TIF)

**S3 Fig. Total counts of all objects (A) and average counts of all objects per model run (B).** Results show for model runs when agents have one of two technology types (overlap is 1). Error bars in panel B represent the 95% confidence intervals on the average per model run. (TIF)

**S4 Fig. Density distributions of average recycling intensity values for each model run by movement and occupational intensity (A) and by selection criteria (B).** Dotted lines show the mean value of the distributions. (TIF)

**S5 Fig. Model trends for recycling intensity and number of recycled objects created per time step for different values of scavenging probability and blank probability, split by object type preferences (A & B), size preferences (C & D), and strict/non-strict selection (E & F).** Trend lines for model runs with an overlap parameter of 1 and 100 agents. (TIF)

**S6 Fig. Trends in the behavioral events over the course of model by overlap parameter.** Behavioral events include: number of blanks produced (A), number of discard events (B), number of scavenging events, and number of artifact retouches (D). (TIF)

**S7 Fig. Regression coefficient values for linear regressions run on coefficients of variation (COV) of recycling intensity, flake counts, nodule counts, discard events, scavenging events, number of retouches, and number of encounters for each grid square.** Each facet shows the effect of a different dependent variable on the COVs of each output. (TIF)

**S8 Fig. Estimated effects for blank probability and scavenging probability on coefficients of variation (COV) of recycling intensity, flake counts, nodule counts, discard events, scavenging events, number of retouches, and number of encounters for each grid square.**

Estimated effects calculated from regression coefficients for blank probability, scavenging probability, and interaction terms between the two.
(TIF)

**S9 Fig. Boxplots of coefficients of variation (COV) of recycling intensity across all grid squares from each model run.** Only data from model runs where agents had one of two technology types (overlap = 1) and 200 agents occupied the landscape during a model run.
(TIF)

**S10 Fig. Boxplots of coefficients of variation (COV) of recycling intensity across all grid squares from each model run.** Only data from model runs where agents had one of two technology types (overlap = 1), 200 agents occupied the landscape during a model run, and flakes are preferred.
(TIF)

**S11 Fig. Correlations with recycling intensity measured for each grid square displayed by selection parameters.** The correlations are between recycling intensity and: number of discard events (A), number of scavenging events (B), number of grid square encounters (C) and number of retouch events (D).
(TIF)

**S12 Fig. Correlations with recycling intensity measured for each grid square displayed by frequency of recycling behaviors.** Results shown for model runs when the overlap parameter is 1. The correlations are between recycling intensity and: number of discard events (A), number of scavenging events (B), number of grid square encounters (C) and number of retouch events (D).
(TIF)

**S13 Fig. Relationship between recycling intensity values and logged artifact counts for each grid square by recycling frequency.** Results for model runs where agents have one of two technology types (overlap is 1) and only 200 agents occupy the landscape during model run. Linear relationship shown for each mu value. Equation and R squared value for each line given in upper left corner of each panel.
(TIF)

**S14 Fig. Relationship between recycling intensity values and logged artifact counts for each grid square by selection criteria.** Results for model runs where agents have one of two technology types (overlap is 1). Linear relationship shown by dark blue line. Equation and R squared value for each line given in upper left corner of each panel.
(TIF)

**S15 Fig. Distributions of assemblage size for each grid square.** Results shown for model runs where agents have one of two technology types (overlap is 1).
(TIF)

**S16 Fig. Cortex ratios averaged over the entire landscape at the end of model run.**
(TIF)

**S17 Fig. Relationship between assemblage density and cortex ratios for each grid square.** Results for model runs where agents have one of two technology types (overlap is 1) and prefer flakes (flake_preference is TRUE).
(TIF)

**S18 Fig. Distributions of number of grid squares identified as high-value hotspots for output variables across all model runs and parameter sets.**
(TIF)

**S19 Fig. Correlations between retouched artifact proportions and grid square encounters measured for each grid square.**
(TIF)

**S20 Fig. Coefficient of variation (COV) for recycling intensity through time across all model runs.**
(TIF)

**S21 Fig. Correlations between A) object counts and discard events, B) object counts and scavenging events, C) object counts and encounters with layers, D) object counts and retouches, E) discard events and scavenging events, F) discard events and encounter with layers, G) discard events and retouches, H) scavenging events and retouches, I) scavenging events and encounters with layers, and J) encounters with layers and retouches.** Correlations are calculated for each grid square.
(TIF)

**S22 Fig. Coefficient of variation (COV) for cortex ratios across all model runs.**
(TIF)

## Acknowledgments

The author would like to thank Radu Iovita and Colin Wren for their helpful comments on the first draft of this paper. The author would also like to thank the two reviewers whose comments greatly improved the quality of this manuscript. This work also made use of the NYU IT High Performance Computer resources and services.

## Author Contributions

**Conceptualization:** Emily Coco.

**Data curation:** Emily Coco.

**Formal analysis:** Emily Coco.

**Funding acquisition:** Emily Coco.

**Investigation:** Emily Coco.

**Methodology:** Emily Coco.

**Project administration:** Emily Coco.

**Resources:** Emily Coco.

**Software:** Emily Coco.

**Supervision:** Emily Coco.

**Validation:** Emily Coco.

**Visualization:** Emily Coco.

**Writing – original draft:** Emily Coco.

**Writing – review & editing:** Emily Coco.

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
