## [Decision Letter · Decision Letter 0]

22 Sep 2023

PONE-D-23-27739Exploring the impact of mobility and selection on stone tool recycling behaviors through agent-based simulationPLOS ONE

Dear Dr. Coco,

Thank you for submitting your manuscript to PLOS ONE. After careful consideration, we feel that it has merit but does not fully meet PLOS ONE’s publication criteria as it currently stands. Therefore, we invite you to submit a revised version of the manuscript that addresses the points raised during the review process.

Both reviews are rather positive. Please do your best to adress the remarks and recommendations

We look forward to receiving your revised manuscript.

Kind regards,

Ran Barkai

Academic Editor

PLOS ONE

[The author would like to thank Radu Iovita and Colin Wren for their helpful comments on the first draft of this paper. This research is based on work supported by the National Science Foundation under Grant No. 2133751. Funding for this work was also provided by the Leakey Foundation. This work was also supported in part through the NYU IT High Performance Computer resources and services.]

 [EC was supported by the National Science Foundation under Grant No. 2133751 and  by the Leakey Foundation. The funders had no role in study design, data collection and analysis, decision to publish, or preparation of the manuscript.]

Additional Editor Comments:

Both reviews are rather positive. Please do your best to adress the remarks and recommendations

Reviewers' comments:

Reviewer's Responses to Questions

**Comments to the Author**

1. Is the manuscript technically sound, and do the data support the conclusions?

Reviewer #1: Yes

Reviewer #2: Yes

2. Has the statistical analysis been performed appropriately and rigorously? 

Reviewer #1: Yes

Reviewer #2: Yes

3. Have the authors made all data underlying the findings in their manuscript fully available?

Reviewer #1: Yes

Reviewer #2: Yes

4. Is the manuscript presented in an intelligible fashion and written in standard English?

Reviewer #1: Yes

Reviewer #2: Yes

5. Review Comments to the Author

Reviewer #1: For this review, as is my general policy for reviewing junior scholars’ work, I waive my anonymity – hello, this is Ellery Frahm, and I’ve done ABM of inter-group contacts during lithic procurement activities.

This is a well-written manuscript that considers an important question and applies ABM appropriately as a means to identify significant variables with respect to the issue of lithic recycling.

Other reviewers might suggest some other variables or conditions that they would want to see in the model, but I consider the model useful as it is. Were there other variables that I would like to have seen? Sure, and the major one would be some sort of sedimentation rate proxy that takes slowly older lithics out-of-play. However, having done ABM, I know that any sort of model is a simplified cartoon of reality, and a modeler must decide, especially during the process of testing variables and identifying key ones, what those simplifications are.

Other reviewers might offer a critique that the ABM results are commonsensical or even obvious, like the outcome that basically states there are more recycled lithics when there is more recycling behavior. I am also fine with this aspect of the manuscript. It is important to test ideas that even seem “obvious” on the surface. And sometimes it is not only the qualitative result but also the magnitude of the effect that is an important finding.

My only suggestion for revision concerns the discussion of mu (which, to be pedantic, would ideally be the Greek letter in the text as well). There are occasional points that contrast when mu is high versus when mu is low (Lines 974-976 as one example), which leads the reader to flip back in the manuscript to re-acquaint themselves with what mu was and think again about what it means for the past – and if they find Table 2 first, then they spend time trying to think again about how low vs high movement redundancy translates to hunter-gatherer behavior. I would encourage the author to think about what h-g behaviors, as discussed (or not) in the literature, correspond to movement redundancy on the landscape and to (1) both explicitly discuss that and (2) discuss the outcomes of varying mu values in those terms. That way, the discussion would read less like “When mu is high…” and more like “When Phenomenon X is more common…,” emphasizing interest in behaviors rather than equations.

In short, there is a lot to like about this manuscript, and I’m still digesting what some of it means and what all of the implications are, but more importantly, it is important to have a way to think about this issues in a novel way.

Reviewer #2: PONE-D-23-27739, “Exploring the impact of mobility and selection on stone tool recycling behaviors through agent-based simulation” by Coco.

Please note:

1. I consulted none of the supplementary materials.

2. Anonymity waived. This is M. Shott.

This paper follows from, but significantly extends the Coco et al. (2021) paper on which it builds. It does this by a more sophisticated modeling of mobility and by including selection criteria for scavenging. In this way, it moves the research line forward. The paper also demonstrates, if further demonstration is needed, the value of landscape or distributional vs. “site” approaches to the understanding of hunter-gatherer archaeological records. I recommend publication even in the paper’s current form, but also have a few suggestions that might improve it modestly.

Undeniably, the paper bears an Old World Paleolithic (OWP) cast, in its focus upon cores and flakes, implicitly in its focus on cores and retouched flakes rather than formal tools (e.g. points, scrapers, that are more common in the Americas), the central importance of cortex ratio as analytical device, consideration of technologies and “transitional” assemblages and, of course, in its legitimate concern with long-term tool exposure and time frames that are more geological or truly archaeological than shorter time frames. This is understandable, and not a criticism. However, the paper somewhat imperfectly accesses relevant North American literature (i.e. not empirical North American studies but more theoretical work by North Americanists.)

The paper is written fairly well but is remarkably dense in substantive terms, less a criticism than an observation. Certainly it is systematic in defining its problem, designing simulations that can test common assumptions about the meaning of tool and assemblage variation and in reporting of results. Indeed, the latter are given in copious detail that catalogues the complex covariation among variables. I mean this as praise, but it also makes the paper a challenge to read and assimilate. Varying clarity of prose also handicaps the paper’s value. For instance, around lines 875-8, Coco writes, “recycling intensity, which is measured as the ratio of recycled objects to total objects in an assemblage, increases on a landscape scale when there is a higher probability that agents are practicing scavenging behaviors and when agents are more frequently retouching flakes.” Her analysis shows that Coco’s conclusions are not tautological, but this statement invites that misinterpretation. Since scavenging, in this paper’s context, amounts to recycling, and retouch of scavenged/recycled objects amounts to, well, retouch, here she seems to say “recycling [as incidence, proportion or “intensity”] increases when recycling increases and retouch increases when retouch increases.” I recommend rewording here to clarify the paper’s results.

There is also the matter of equifinality, briefly acknowledged around Line 1011. Relationships between tools properties (size, technological origin [i.e. flake vs. nodule/core], incidence of retouch [how many tools among all tools], original vs. discard location) and assemblages (“sites”) and landscapes are quite complex, as the paper shows. Clearly, the paper has important implications (e.g., for the incidence of recycled tools at hotspots qua sites). But so many outcomes contingent upon so many starting conditions will make it difficult to apply Coco’s simulation results to empirical cases. Or rather, it makes the model almost infinitely protean, capable of being used selectively or partially to explain any patterns observed in empirical data. Complexities of equifinality and model specifications are addressed somewhat from Lines 1010-30. She might reflect on and then advise others somewhat more thoroughly about how to address the equifinality challenge in so complex a setting of variables and their interactions.

One grid-square variable is called “reduction intensity,” by which Coco means the proportion of tools among all tools. This wording is unfortunate, because it confuses recycling incidence (count) and proportion (ratio of recycled tools/all tools) with what many archaeologists, I believe, mean by “intensity” either of reduction or recycling, i.e. how much or to what extent each tool is reduced. This understanding of intensity, of course, relates to the curation concept when it’s defined as the ratio of realized to maximum utility. I strongly urge Coco to reconsider use of “intensity” to mean count or % of retouched tools, and instead to call it recycling “incidence” or “proportion.” These terms are both more accurate and recognize that “intensity” often—usually?—pertains to degree of retouch of individual tools.

On a related matter, one element omitted from the research design is degree or extent of recycling, which amounts to extent of reduction, i.e., what I’d call, contra Coco, reduction intensity. This is understandable because the behavior of so many other variables is modeled, and one more might greatly complicate analysis, interpretation and reporting of results. But extent of curation is important both empirically (notably in Dibble’s original reduction thesis, itself imported to the Americas in Hoffman’s mid-1980s paper) and in theory or concept, where—e.g. in my 1996 Jnl. of Anthr. Res. paper—it is defined as a ratio between realized and maximum utility. Granted, in Coco’s OW Paleolithic context, curation so defined requires further specification owing to possible differences in maximum utility depending upon raw-material supply and technology over geological time. That is, what was “used up” in, say the Middle Pal. might be considered still useful in the Upper Pal. But the modeling described here treats recycling as a categorical variable (i.e. either a tool was recycled or it wasn’t) when the curation that underlies it actually is a continuous one. I don’t know but wonder if categorical “recycling” affects model results. But I suspect as much when results are expressed as “intensity” (again, incidence or proportion) of recycled tools compared to landscape variables. If “intensity” is not mere incidence, but considered as varying amount of reduction per categorically recycled tool, then model results for recycled-tool incidence/proportion will differ from those reported here if degree of reduction—again, curation—varies both significantly and continuously among tools. Two tools can be equally recycled, one of them slightly and the other greatly. Collapsing this variation into “presence of recycling” seems a serious model limitation. I hope that Coco will integrate the continuous nature of tool reduction (i.e., extent of reduction) in future modeling. Somewhat obliquely, this matter is addressed ca. Line 910 where Coco briefly acknowledges the effect of use-life as a limited, not infinite, quantum. But that passage essentially says “I haven’t considered this [yet?].” That’s fine, but she might do more than that, for instance by considering how use-life and the linked (but not identical) process of curation can directly affect her model results.

“Curation” occurs only once in the text where, like the titles of two cited papers in which it also appears, it passes undefined. The text occurrence might imply some equivalence between curation and “technological efficiency,” also undefined. Coco’s citations 63-68 are merely a selection, of course, and not necessarily the most representative ones, of the curation issue. Indeed, the first of these citations—Bamforth, mid-1980s—is a conceptually unclear paper that, among other things, inspired my efforts to try to clarify both the curation concept and the technological-organization approach that Bamforth misunderstood. This is no mere semantic quibble. Conceptually, recycling is subsumed by curation, which makes the latter directly relevant here. At the same time, Lines 420-30 here caused me to usefully rethink the curation concept in the context of Dibble’s reduction thesis, as a process that unfolds not only—or at times perhaps even mostly—in ecological or real-time but geological or archaeological chronological scales as well.

Accordingly, like the Barton and Riel-Salvatore paper that it cites, this ms. has, from an American perspective, a highly constrained view of assemblage variation, limiting it essentially to variation by the proportion or incidence of retouched/recycled tools vs. nodules and, presumably, debris. Again, that quality reflects its OWP cast, but one useful direction for future research is to define assemblage variation more broadly, both in terms of number of “types,” (sidestepping for the moment the validity or integrity of types, especially as defined in OW Middle Paleolithic context and which, anyway, Dibble’s thesis somewhat reconceptualized as various tool “types” representing progressive stages in the degree and pattern of retouch of original flake blanks/tools) and the size-composition relationships that govern assemblage accumulations (sensu my 2010 paper and Phillips et al.’s recent Asian Perspectives paper; for instance, Lines 234-5 make a similar point more extensively elaborated in mine of 2010). One possibility, as above, reconceives the many Middle Paleolithic Bordean “types,” for instance, as Dibble did: size-shape transformations of varying type and degree via reduction of a few basic technological flake-tool types. In this context as in others (again, my 2010 and Phillipps et al. recent paper), assemblage composition as approximated by proportions of Bordean “types” patterns clearly with assemblage size as a function perhaps of occupation span or intensity (as detailed in my 2003 chapter in Moloney and Shott, that no one ever read). Coco’s paper documents similar correlations between assemblage properties and patterns of land-use that might be more broadly contextualized by considering other approaches cited in this paragraph.

Coco’s discussion around Lines 90-100 essentially is what North American scholars call “niche-construction theory” (e.g. in her sources 32 and 66). As a substantive matter, that’s neither here nor there, but Coco might acknowledge this parallel so as to clarify the conceptual similarities in her and their approaches. Continuing, the discussion on Lines 99-110 or so, greatly elaborated upon later, put me in mind of Surovell’s model (in his 2012 book) that related occupation span to the ratio between two quantities: formal tools made of nonlocal cherts to used flakes made of local cherts. Like hers, Surovell’s model is simplified as it must be; simulation always identifies key variables and charts their covariation under various conditions, not recreate the past in all details. But Coco’s model seems to build on, even surpass Surovell’s in its greater complexities of land-use and “site” occupation and in consideration of repeated, if intermittent, use of flakes and/or nodules. Still, citing Surovell’s model better contextualizes Coco’s own work.

The Discussion section is unusually long and somewhat repetitive. It might be revised certainly by editing and shortening, and perhaps by reorganization as numbered sections that each address one of the various points made in the results section and repeated there.

The text is clean, with only two copy-editing glitches that I caught:

Line 32: “those that [have] been exposed”

Line 64: “the productions and flakes” [the production of flakes??]

6. PLOS authors have the option to publish the peer review history of their article (what does this mean?). If published, this will include your full peer review and any attached files.

Reviewer #1: No

Reviewer #2: **Yes: **Michael J. Shott

---

## [Author Response · Author response to Decision Letter 0]

26 Oct 2023

Please see attached Response to Reviewers document to see point by point response. I would like to thank both the reviewers and the editor for their helpful feedback.

---

## [Editor Report · Decision Letter 1]

27 Oct 2023

Exploring the impact of mobility and selection on stone tool recycling behaviors through agent-based simulation

PONE-D-23-27739R1

Dear Dr. Coco,

We’re pleased to inform you that your manuscript has been judged scientifically suitable for publication and will be formally accepted for publication once it meets all outstanding technical requirements.

Kind regards,

Ran Barkai

Academic Editor

PLOS ONE
---

## [Editor Report · Acceptance letter]

31 Oct 2023

PONE-D-23-27739R1 

Exploring the impact of mobility and selection on stone tool recycling behaviors through agent-based simulation 

Dear Dr. Coco:

I'm pleased to inform you that your manuscript has been deemed suitable for publication in PLOS ONE. Congratulations! Your manuscript is now with our production department. 

Kind regards, 

on behalf of

Professor Ran Barkai 

Academic Editor

PLOS ONE